# Lithium tantalate photonic integrated circuits for volume manufacturing

Chengli Wang[1,2,3,4], Zihan Li[2,3,4], Johann Riemensberger[2,3], Grigory Lihachev[2,3], Mikhail Churaev[2,3], Wil Kao[2,3], Xinru Ji[2,3], Junyin Zhang[2,3], Terence Blesin[2,3], Alisa Davydova[2,3], Yang Chen[1], Kai Huang[1], Xi Wang[1], Xin Ou[1✉] & Tobias J. Kippenberg[2,3✉]

Electro-optical photonic integrated circuits (PICs) based on lithium niobate (LiNbO$_3$) have demonstrated the vast capabilities of materials with a high Pockels coefficient[1,2]. They enable linear and high-speed modulators operating at complementary metal–oxide–semiconductor voltage levels[3] to be used in applications including data-centre communications[4], high-performance computing and photonic accelerators for AI[5]. However, industrial use of this technology is hindered by the high cost per wafer and the limited wafer size. The high cost results from the lack of existing high-volume applications in other domains of the sort that accelerated the adoption of silicon-on-insulator (SOI) photonics, which was driven by vast investment in microelectronics. Here we report low-loss PICs made of lithium tantalate (LiTaO$_3$), a material that has already been adopted commercially for 5G radiofrequency filters[6] and therefore enables scalable manufacturing at low cost, and it has equal, and in some cases superior, properties to LiNbO$_3$. We show that LiTaO$_3$ can be etched to create low-loss (5.6 dB m$^{-1}$) PICs using a deep ultraviolet (DUV) stepper-based manufacturing process[7]. We demonstrate a LiTaO$_3$ Mach–Zehnder modulator (MZM) with a half-wave voltage–length product of 1.9 V cm and an electro-optic bandwidth of up to 40 GHz. In comparison with LiNbO$_3$, LiTaO$_3$ exhibits a much lower birefringence, enabling high-density circuits and broadband operation over all telecommunication bands. Moreover, the platform supports the generation of soliton microcombs. Our work paves the way for the scalable manufacture of low-cost and large-volume next-generation electro-optical PICs.

Next-generation ultrahigh-speed PICs based on electro-optical materials are poised to play a role in energy-efficient data centres, optical communications, 5G and 6G radiofrequency filters and in particular in AI workload-driven high-performance computing, provided that scalable low-cost manufacturing becomes possible. In the past two decades, PICs based on silicon (silicon photonics) have rapidly transitioned from academic research to widespread use in telecommunications[8] and data centres[9]. One crucial factor driving the commercial feasibility of this technological revolution is the high-volume availability and cost-effectiveness of SOI wafers. These SOI wafers, prepared using smart-cut (ion slicing) techniques[10], enable the manufacture of silicon photonics but crucially are more widely used in consumer microelectronics. Globally, more than 3 million SOI wafers are produced each year, with the wafer diameter being as large as 300 mm[8]. Using a similar technique, LiNbO$_3$ has been fabricated into lithium niobate-on-insulator (LNOI) structures, offering an entirely new class of ultrahigh-speed, low-voltage electro-optical PICs[3,11,12] that can become key components in future energy-efficient communication systems. Despite the tremendous scientific progress and the

increased application range of LiNbO$_3$ PICs, the path to commercialization remains difficult. Unlike SOI technology, LNOI lacks a larger volume of consumer electronics driving its demand, resulting in economic limitations to its commercialization. By contrast, another ferroelectric material, LiTaO$_3$, which has similar structural properties to LiNbO$_3$, has entered the large-volume production stage, driven by its applications in 5G filters[13,14], and is projected to achieve a production capacity of 750,000 lithium tantalate-on-insulator (LTOI) wafers a year by 2024 (ref. 15). This substantial volume enables considerable benefits in terms of low-cost production when adopting LTOI as a platform for PICs, but PICs based on this material have not been reported to date. LiTaO$_3$, as well as having a large production volume, exhibits comparable, or in some cases superior, properties to LiNbO$_3$. LiTaO$_3$ is an oxygen octahedral ferroelectric crystal with a crystal structure that is nearly identical to that of LiNbO$_3$, replacing Nb atoms with the heavier Ta atoms. This change gives LiTaO$_3$ not only a greater mass density but also stronger chemical bonds, resulting in increased strength and chemical stability[16]. The optical bandgap of LiTaO$_3$ (3.93 eV) is larger than that of LiNbO$_3$ (3.78 eV)[17–19], enabling nonlinear optical conversion

[1]National Key Laboratory of Materials for Integrated Circuits, Shanghai Institute of Microsystem and Information Technology, Chinese Academy of Sciences, Shanghai, China. [2]Institute of Physics, Swiss Federal Institute of Technology Lausanne, EPFL, Lausanne, Switzerland. [3]Center of Quantum Science and Engineering, EPFL, Lausanne, Switzerland. [4]These authors contributed equally: Chengli Wang, Zihan Li. ✉e-mail: ouxin@mail.sim.ac.cn; tobias.kippenberg@epfl.ch

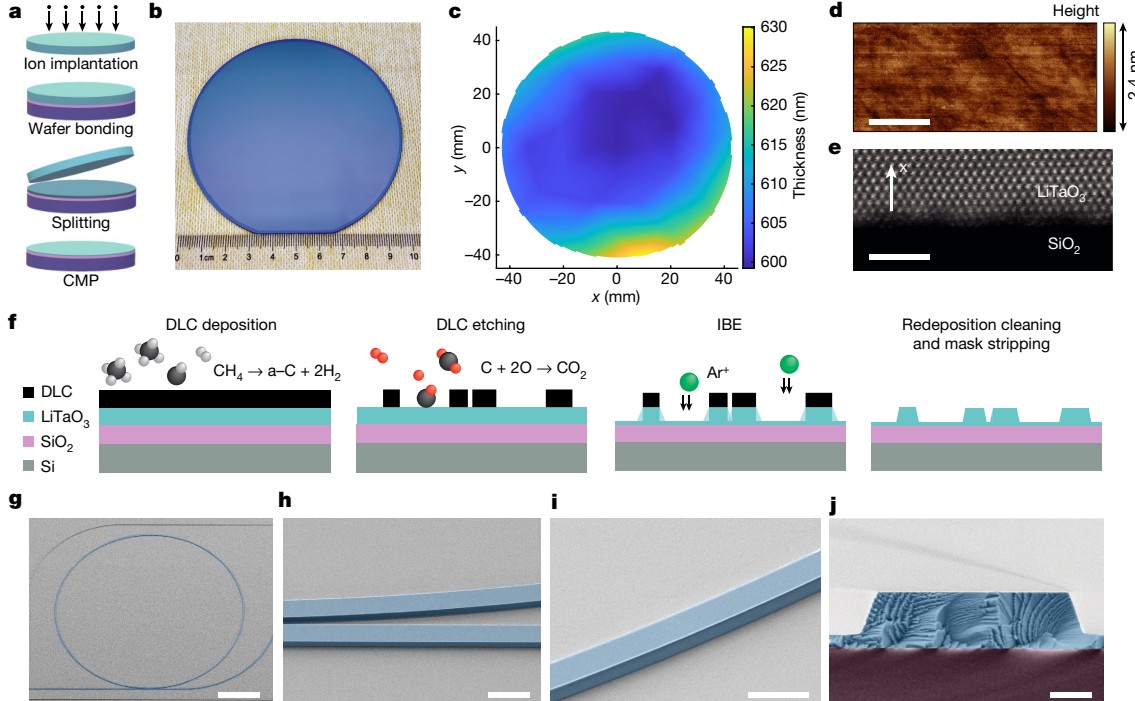

**Fig. 1 | LTOI substrates and optical waveguides. a**, Schematic of the LTOI wafer-bonding workflow showing hydrogen-ion implantation, bonding, splitting and chemical mechanical polishing (CMP). **b**, Photograph of the bonded wafer demonstrating uniform and defect-free bonding. **c**, Thickness map of the $LiTaO_3$ thin film on the wafer. The $x,y$ axes represent the distance from the wafer centre. **d**, Atomic force micrograph of the $LiTaO_3$ thin film surface. Scale bar, 500 nm. **e**, High-resolution scanning transmission electron-microscopy image of the $LiTaO_3$–$SiO_2$ bonding interface. The arrow represents the x-cut crystal orientation. Scale bar, 2 nm. **f**, Schematic of the fabrication workflow for LTOI optical waveguides, including DLC hard-mask

deposition by plasma-enhanced chemical vapour deposition (PECVD) from the methane precursor, DLC dry etching through oxygen plasma, and $LiTaO_3$ etching by argon ion-beam etching (IBE), followed by redeposition and mask removal. The layers are DLC (black), $LiTaO_3$ (light blue), $SiO_2$ (purple) and Si (grey). Spheres show C (black), O (red) and $Ar^+$ (green). **g**, Colourized scanning electron micrograph (SEM) of LTOI microring resonator (blue). Scale bar, 50 μm. **h**, Colourized SEM of etched LTOI microring and bus waveguide coupling section. Scale bar, 2 μm. **i**, Colourized SEM of etched LTOI waveguide and sidewall. Scale bar, 2 μm. **j**, Colourized SEM cross-section of etched LTOI waveguide (blue) on top of $SiO_2$ bottom cladding (purple). Scale bar, 500 nm.

to the visible and even ultraviolet[20]-wavelength range. Furthermore, the material exhibits a greatly decreased optical anisotropy, that is, the magnitude of the optical birefringence is reduced more than 10-fold compared with $LiNbO_3$, which suppresses mode mixing, as can occur in tight waveguide bends. Moreover, $LiTaO_3$ features a similar Pockels coefficient ($r_{33}$ = 30.5 pm V$^{-1}$) to the well-established $LiNbO_3$ ($r_{33}$ = 30.9 pm V$^{-1}$) with a moderately larger electrical permittivity, $\epsilon_{33}$ = 43, implying that the modulation efficiency of the two materials is expected to be almost identical. Furthermore, $LiTaO_3$ benefits from a larger optical damage threshold, which is relevant for high-power applications. Of particular relevance for applications in the realm of microwave-optical quantum transduction[21,22], the nearly 10-fold-lower microwave loss tangent of $LiTaO_3$ (refs. 23,24) is a promising avenue to improve device performance to unity conversion efficiency, which has so far eluded efforts using $LiNbO_3$ owing to the limited quality factors of microwave resonators[22]. Historically, despite the beneficial optical material properties, the use of $LiTaO_3$ for optical modulators in optical communication networks has been limited. One of the reasons is that the Curie temperature of $LiTaO_3$ (610–700 °C, depending on the Li:Ta ratio) is much lower than the temperature needed for the fabrication of optical waveguides by ion diffusion (typically more than 1,000 °C), which compounded the use of $LiTaO_3$ for bulk modulators on the basis of the ion diffused waveguide[25]. For this reason, legacy bulk modulator technology has used $LiNbO_3$. However, the commercial use of LTOI in wireless applications, owing to its suitable acoustic properties, combined with the above optical properties, makes it an ideal platform for scalable manufactured electro-optical PICs, although such a use has never been demonstrated or pursued. Although free-standing

'whispering gallery' mode resonators have been fabricated from $LiTaO_3$ single crystals[26], as a result of femtosecond laser direct writing[27] or focused ion beam milling[28], scalable manufactured PICs have remained an outstanding challenge.

Here, we overcome this challenge and implement what is to our knowledge the first PIC platform using LTOI based on direct etching[7], and demonstrate ultralow optical loss, electro-optical tuning, switching through the Pockels effect and soliton-microcomb generation through the optical Kerr effect of $LiTaO_3$. We achieve this by transferring the diamond-like carbon (DLC)-based masking etching process, originally developed for $LiNbO_3$, to $LiTaO_3$, and propose a new solution to remove $LiTaO_3$ redeposition, which highlights the flexibility of our process for the fabrication of a variety of ferroelectric photonics platforms. We also demonstrate a DUV approach to electrode manufacturing. Taken together, our work establishes a basis for scalable volume manufacturing of ultrahigh-speed electro-optical PICs.

## $LiTaO_3$ PICs

The fabrication process for LTOI wafers and optical waveguides is depicted in Fig. 1 (details in Methods). The LTOI wafer was fabricated by the smart-cut technique[14]. The process flow is schematically illustrated in Fig. 1a. In contrast to the well-established LNOI preparation process, which uses helium-ion implantation with an energy greater than 200 keV[29], the fabrication of LTOI favours hydrogen ions with an implanted energy of 100 keV and a beam current ten times higher, as found in most commercial ion implanters, thereby simplifying the wafer production. The fabrication recipes of LTOI are more closely

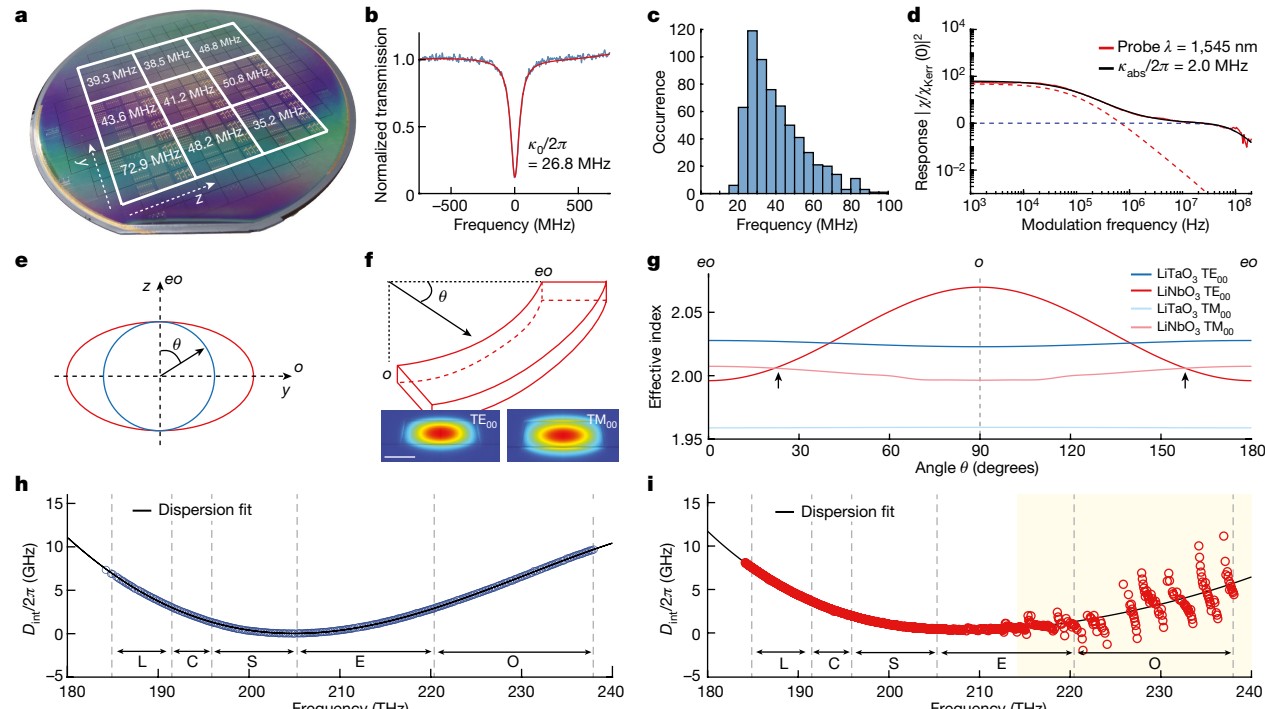

**Fig. 2 | Optical characterization of LTOI PICs. a**, Wafer-scale map of mean intrinsic loss, $\kappa_0/2\pi$, for similar resonators. **b**, Normalized resonance transmission spectrum of an optical racetrack microresonator at 209.358 THz. **c**, Statistical distribution of intrinsic loss, $\kappa_0/2\pi$, of the optical racetrack microresonator. Mean = 40.4 MHz, median = 36.4 MHz. **d**, Nonlinear optical-response measurement (solid red line) and fit (solid black line) of the thermo-optical (red dashed line) and Kerr (blue dashed line) nonlinear responses of the optical microresonator, demonstrating ultralow optical absorption loss. **e**, Illustration of the LiNbO$_3$ (red) strongly negative uniaxial and LiTaO$_3$ (blue) weakly positive uniaxial crystal birefringence. For LiTaO$_3$, ordinary refractive index $n_{LT.o}$ = 2.119, extraordinary refractive index $n_{LT.e}$ = 2.123 and the difference $\Delta n_{LT}$ = 0.004; for LiNbO$_3$, ordinary refractive index $n_{LN.o}$ = 2.21, extraordinary refractive index $n_{LN.e}$ = 2.14 and the difference $\Delta n_{LN}$ = −0.07. **f**, Illustration of the curve angle, $\theta$, and fundamental transverse electric (TE$_{00}$) and transverse

magnetic (TM$_{00}$) mode profiles in LTOI. Scale bar, 1 μm. **g**, Numerical simulation of fundamental TE$_{00}$ and TM$_{00}$ optical-mode effective refractive indices of LNOI (red) and LTOI (blue) as a function of the angle between the waveguide and the *y* axis of the x-cut LNOI or LTOI film. The reduced birefringence of LTOI precludes unwanted birefringent mixing between fundamental TE$_{00}$ or TM$_{00}$ modes in thick waveguides. Arrows indicate birefringent mode mixing. **h**, Dispersion profile of an LTOI racetrack microresonator with the waveguide cross-section 2 μm × 0.5 μm and a slab 100 nm thick. $D_1/2\pi$ = 82.234 GHz, $D_2/2\pi$ = 197.88 kHz. L, C, S, E and O telecommunication bands are marked with vertical dashed lines. **i**, Dispersion profile of an LNOI racetrack microresonator with similar cross-section and strong mode mixing at frequencies above 215 THz, which occupies the E-band and the O-band in the optical communication. $D_1/2\pi$ = 80.83 GHz, $D_2/2\pi$ = 105.72 kHz.

aligned with the high-volume commercial production of SOI wafers, resulting in higher efficiency and lower costs in the production of LTOI than of LNOI. The fabricated LTOI wafer has a 4-inch (102 mm) size with a surface roughness of 0.25 nm and a non-uniformity of less than 30 nm (Fig. 1b–d). The crystallinity of LiTaO$_3$ and the LiTaO$_3$–SiO$_2$ interface remain of high quality after the completion of the production process, and so does the sharpness of the bonding interface, as can be seen in the high-resolution scanning transmission electron microscopy image (Fig. 1e). Photonic building blocks such as optical ring resonators (Fig. 1g), racetrack resonators and waveguide spirals are also fabricated. The lithography, dry etching and by-product cleaning processes were optimized to achieve both favourable coupling regions (Fig. 1h) and well-defined, smooth sidewalls (Fig. 1i) of the LTOI PICs. As detailed in Methods, the removal of non-volatile by-products for LTOI requires a different chemical than for LNOI[7]. The cleaved cross-section featuring steep sidewall angles of almost 70° with respect to the surface is shown in Fig. 1j.

Next, we characterized the LiTaO$_3$ PICs (D101_LT_A2) using frequency-comb calibrated tunable diode laser spectroscopy[30] to determine the optical loss and absorption of optical microresonators with a waveguide width of 2.0 μm across the 4-inch wafers (Fig. 2a). We find mean intrinsic loss rates, $\kappa_0/2\pi$, of between 35.2 MHz and 72.9 MHz with eight of the nine fields performing better than 50.8 MHz. The microresonator intrinsic loss rate $\kappa_0/2\pi$ = 35.2 MHz corresponds to a propagation

loss, $\alpha$, of 7.3 dB m$^{-1}$ for the unreduced LiTaO$_3$ wafer that is used for optical applications. We also characterized the optical loss of the LTOI platform fabricated from the reduced-LiTaO$_3$ bulk wafers, known in the filter industry as acoustic grade or black LiTaO$_3$, which undergo an extra chemical reduction step, typically by annealing in carbon powder to minimize the pyroelectric effect[31] (Methods and Extended Data Fig. 1). The LTOI fabricated from the reduced wafer exhibits an intrinsic loss rate $\kappa_0/2\pi$ = 42 MHz in the best field and a mean value of $\kappa_0/2\pi$ = 82 MHz across the whole wafer. This corresponds to losses $\alpha$ = 8.8 dB m$^{-1}$ and $\alpha$ = 17.1 dB m$^{-1}$, which is below the published losses of the wafer-scale fabrication of LNOI PICs[32], with an average loss of 27 dB m$^{-1}$, making our DUV-based process directly applicable to widely used mass-manufactured LTOI wafer substrates. An optical-resonance transmission spectrum (D101_LT_A2_F1_C4_02_WG4) and fit is shown in Fig. 2b, which indicates an intrinsic loss rate of $\kappa_0/2\pi$ = 26.8 MHz; this corresponds to a propagation loss of $\alpha$ = 5.6 dB m$^{-1}$ for unreduced LiTaO$_3$. We also fabricated optical-waveguide spirals with a waveguide cross-section of 1.75 μm × 0.6 μm and found a propagation loss of around 9 dB m$^{-1}$ (Extended Data Fig. 2). A histogram of fitted intrinsic loss rates for the microresonator is shown in Fig. 2c. The contributions of optical absorption and scattering from bulk and sidewall imperfections can be separated by thermal response spectroscopy[33] (Fig. 2d). An intensity-modulated pump laser was tuned to the centre of the optical resonance, and the frequency-modulation response of the optical

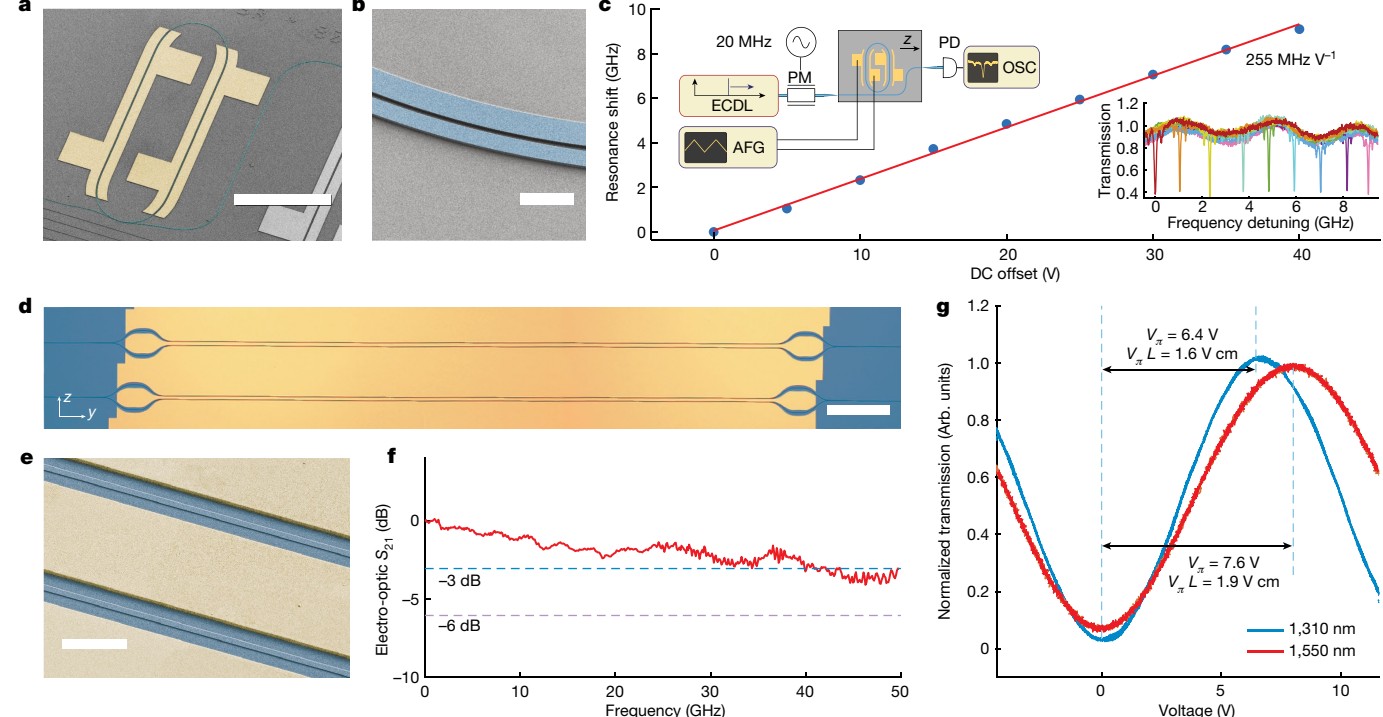

**Fig. 3 | Electro-optical tuning and switching in LTOI. a**, Colourized SEM of LTOI (blue) racetrack optical microresonator with gold electrodes (yellow). Scale bar, 300 µm. **b**, Colourized SEM of pulley resonator and bus waveguide coupling section. Scale bar, 5 µm. **c**, Measured resonance shift as a result of tuning voltage. The linear fit indicates a voltage tuning response of 255 MHz V⁻¹. Left inset: schematic of the measurement set-up for microresonator tuning measurement with phase-modulation (PM) sideband calibration. ECDL, external cavity diode laser; AFG, arbitrary frequency generator; OSC, oscilloscope; PD, fast photodiode. Right inset: normalized cavity transmission measurement showing electro-optical tuning of an LTOI resonance. Each colour step corresponds to an increase in DC tuning voltage of 5 V. **d**, Optical micrograph of 2.5 mm-long MZM. Scale bar, 200 µm. **e**, Colourized SEM of MZM waveguides and electrodes. Scale bar, 10 µm. **f**, Electro-optical bandwidth ($S_{21}$ parameter, measured as the power ratio) of MZM for a device length of 2.5 mm at a wavelength of 1,550 nm. **g**, Normalized optical transmission as a function of applied voltage on travelling-wave electrodes at wavelengths of 1,310 nm and 1,550 nm, showing a voltage–length product, $V_\pi L$, of 1.6 V cm at the O-band and 1.9 V cm at the C-band.

microresonator resulting from the thermo-optical and Kerr effects was read out with a second laser tuned to the side of another resonance. We modelled the frequency dependence of the thermal effect arising from the optical absorption and the optical Kerr effect using finite-element simulations and fitted the combined response[33,34]. We found that the absorption limit of our LTOI microresonator is $\kappa_{abs}/2\pi = 2.0$ MHz, corresponding to an absorption-limited propagation loss of $\alpha = 0.4$ dB m⁻¹, which is close to recent results obtained for LNOI[34]. Therefore, the main source of loss in our tightly confining LiTaO₃ waveguides is scattering losses.

The optical birefringence of LiTaO₃ is more than one order of magnitude smaller than that of LiNbO₃ (Fig. 2e) and therefore enables the fabrication of thick waveguides without incurring mode mixing between the fundamental modes in waveguide bends[7,35]. Mode mixing occurs in x-cut LiNbO₃ waveguide bends when the TE mode transitions from the extraordinary (*eo*) to the ordinary (*o*) axes above a critical LiNbO₃ thickness that at a wavelength of 1.55 µm lies at around 700 nm and at a wavelength of 1.3 µm lies at around 600 nm, largely independently of the slab thickness or waveguide width. In contrast, the low and positive uniaxial birefringence of LiTaO₃ precludes mode mixing in x-cut waveguides with a horizontal-to-vertical aspect ratio greater than one. We simulate the effective mode indices of the fundamental polarization modes of LNOI and LTOI for a waveguide thickness of 600 nm, a waveguide width of 2 µm and a wavelength of 1.55 µm as a function of the angle between the propagation and the *eo* crystal axes (Fig. 2f,g). For LiNbO₃, we found a crossing of the fundamental TE and TM modes at an angle of 25°, whereas no mode crossing is found for an LTOI waveguide with the same dimension. This observation is

in excellent agreement with the results from our optical dispersion measurement, $D_{int} = \omega_\mu - \omega_0 - (D_1 \times \mu)$, where $\mu$ indicates the azimuthal mode index for the mode with frequency $\omega_\mu$, $D_1$ is a free spectral range, $D_{int}$ is an integrated dispersion and $\omega_0/2\pi = 205$ THz, for the LTOI (D101_LT_A2_F1_C4_WG4) and LNOI (D133_02_F2_C5_01_WG4) waveguides, which are depicted in Fig. 2h. The optical microresonators have similar anomalous dispersion, but the dispersion profile of the LTOI microresonator remains smooth over the full measurement span of 185 THz to 240 THz, whereas the LNOI microresonator exhibits striking mode mixing at frequencies above 215 THz. The birefringence complicates the design of compact PICs and is useful only in some special cases, such as birefringence phase matching[36]. Adjustments to the waveguide geometry and working wavelength can weaken the mode mixing caused by strong birefringence in LNOI[7,37], but such adjustments result in reduced optical confinement and chip compactness. By contrast, LTOI offers much lower birefringence, thereby providing greater flexibility in waveguide design and manufacturing, and mode-mixing-free operation over all telecommunications bands from 1,260 nm to 1,625 nm, ranging from the O to the L band. Furthermore, the resonance shift induced by the photorefractive effect in an LTOI microresonator is nearly fivefold smaller than that in an LNOI microresonator (Extended Data Fig. 3 and Methods), which is consistent with the results obtained from bulk crystals[38].

## Electro-optical modulation

To demonstrate the utility of the LTOI platform for electro-optics, we created a tunable high Q-factor microresonator. The resonator

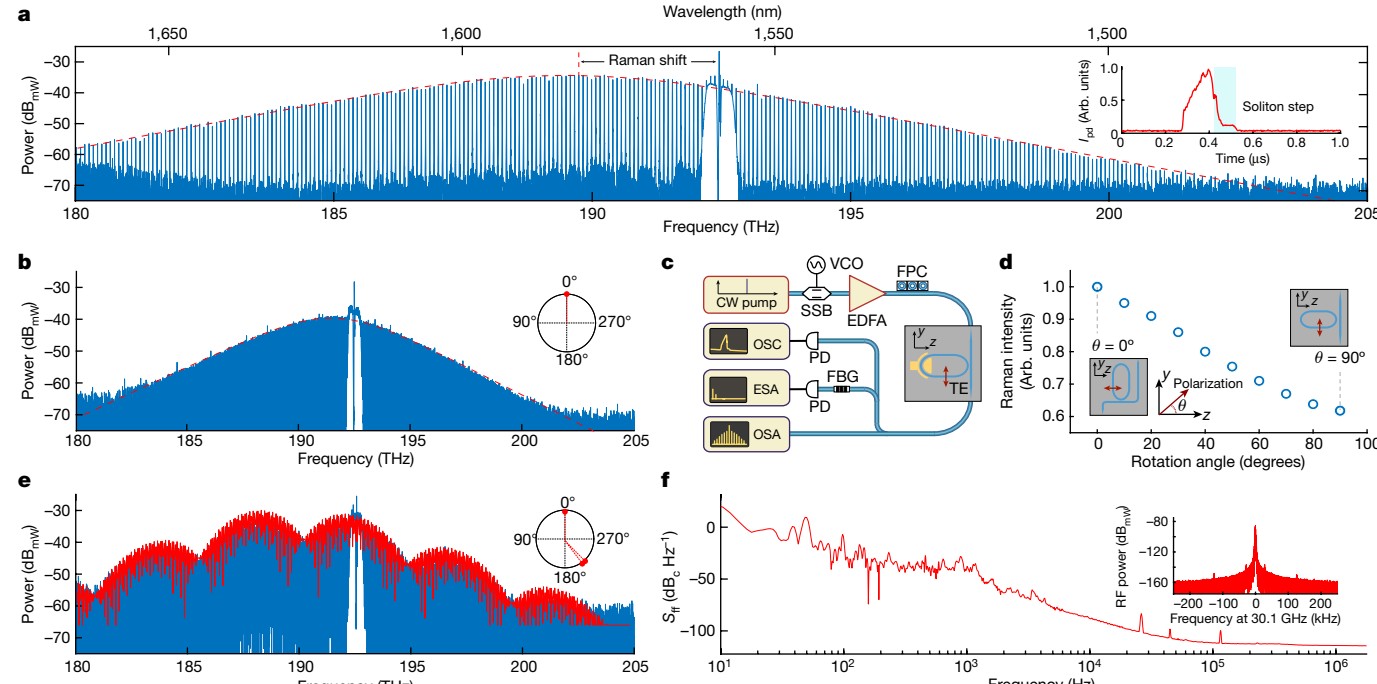

**Fig. 4 | DKS generation in LTOI microresonators. a**, Optical spectrum of a single soliton microcomb featuring a *sech²*-spectral profile with a 3 dB bandwidth of 4.9 THz, corresponding to an FWHM pulse duration of 63 fs at a pulse repetition rate, $f_{rep}$, of 81 GHz. Inset, the light generated during the rapid laser scan measured by filtering out the pump light. The soliton step is marked with light blue shading. $I_{pd}$, photodiode current. **b**, Optical spectrum of a single soliton with a repetition rate of 30.1 GHz; $\Delta T_{FWHM} = 71$ fs. Inset, the relative phase position inside the microresonator. **c**, Optical setup for soliton generation in x-cut LTOI microresonators. The orientation and TE polarization are also indicated in the schematic diagram of the fabricated LTOI chip. Rapid laser scans were generated using a single-sideband modulator (SSB) and voltage-controlled oscillator (VCO), continuous wave (CW) laser and an erbium-doped fibre

amplifier (EDFA). FBG, fibre Bragg grating; FPC, fibre polarization controller. The soliton microcombs were analysed using an optical spectrum analyser (OSA) and the nonlinearly generated light and microwave beat notes were recorded with a PD and analysed with an OSC and electrical spectrum analyser (ESA), respectively. **d**, Variation of the Raman intensity with different LTOI crystal rotation angles. Inset, the angle between the orientation (z axis) and the polarization of the excitation laser. **e**, Optical spectrum of a three-soliton state with a repetition rate of 30.1 GHz. Inset, three solitons inside the microresonator. $\Delta T_{FWHM} = 52$ fs. **f**, Single-side band-phase noise-power spectral density ($S_{ff}$) of a 30.1 GHz microwave beat note generated from the multisoliton state in **e**. Inset, spectrum of a microwave beat note with a resolution bandwidth of 30 Hz.

has a racetrack design with an apex radius of 100 μm and a straight section length of 400 μm (Fig. 3a) with a uniform waveguide width of 2 μm and pulley-style coupling sections (Fig. 3b). Metal electrodes were fabricated using a DUV-lithography-based lift-off process that allowed us to manufacture electrodes with an alignment tolerance of less than 100 nm to the optical waveguide (Methods). We applied a voltage across two of the four electrodes to measure the voltage tuning coefficient and measure the resonance position using an external-cavity diode laser (ECDL) (Fig. 3c and Methods). We found a voltage tuning efficiency of 255 MHz V⁻¹ using a single electrode pair, which corresponds to 510 MHz V⁻¹ if both phase-shifter sections are modulated. We also fabricated a travelling-wave MZM composed of two 50:50 adiabatic Y-splitters at either end and a push–pull optical waveguide phase-shifter pair with a length of 2.5 mm (Fig. 3d). The waveguide width was 1.2 μm and the gap between the LiTaO₃ waveguide sidewalls and the gold electrode was 2.5 μm on each side (Fig. 3e). The etching depth was 220 nm, leaving a 400 nm slab. This maintained a consistent group refractive index, $n_g$, of around 2.25 for both microwave and optics waves. The transmission through the MZM with a 10 kHz triangular voltage sweep is plotted in Fig. 3g. The MZM can work at two communication bands of 1,310 nm and 1,550 nm simultaneously, owing to the use of the broadband adiabatic Y-splitter. The measured $V_\pi$ was 6.4 V for 1,310 nm and 7.6 V for 1,550 nm, corresponding to a $V_\pi L$ of 1.6 V cm and 1.9 V cm, respectively. The difference in $V_\pi L$ arises mainly from the overlap difference between the optical modes and the electric field at distinct wavelengths and increased optical frequency. The measured $V_\pi L$ of 1.9 V cm is similar to the state-of-the-art results

for LNOI at 1,550 nm[1,3], with similar electrode structures, as expected given that LiNbO₃ and LiTaO₃ have almost identical Pockels coefficients (Extended Data Table 1).

We then characterized the small-signal electro-optic bandwidth of the fabricated devices (Methods). The measured 3 dB electro-optic bandwidth was more than 40 GHz (Fig. 3(f)).

## Soliton-microcomb generation

Finally, we investigated the LiTaO₃ platform for soliton-microcomb generation. The strong optical confinement, high Q-factor, anomalous dispersion and substantial Kerr nonlinearity of our LTOI microresonators make them naturally suitable for dissipative Kerr soliton (DKS) generation[39,40]. However, LiTaO₃ is recognized as a Raman-active crystalline material, displaying multiple robust vibrational phonon branches in various polarization configurations[41], which can have a detrimental effect on soliton generation. This Raman interference presents a common challenge when attempting to generate solitons in ferroelectric crystal platforms. For instance, despite extensive research efforts, achieving solitons in the x-cut configuration of LNOI has remained elusive[1,42]. It is well known that the Raman effect is polarization dependent, typically exhibiting maximum strength when the pump light is polarized along the polar axis of the crystal[1,17]. We investigated such a polarization-dependent Raman effect in both x-cut LNOI and LTOI (Methods). A reduction in Raman intensity was achieved when the polarization of incident light transitions from being parallel to the y axis to being parallel to the z axis (the rotation angle θ changes from

0° to 90°; Fig. 4d). We therefore used racetrack microresonators with the straight waveguide section oriented along the $z$ axis ($\theta = 90°$) to minimize the Raman interference. This configuration ensures that the TE mode predominantly aligns with the non-polar $y$ axis, as depicted in the schematic chip diagram in Fig. 4c,d. We used the rapid single sideband tuning scheme described in ref. 43 to overcome thermal non-linearities and initiate solitons at a pump power of 90 mW on-chip using ECDL and an erbium-doped fibre amplifier for pumping. The optical set-up for single-soliton generation is depicted in Fig. 4c (and described in Methods). We achieved single-soliton generation at pulse repetition rates of 81 GHz (D101_LT_A2_F9_C4_02_WG6, Fig. 4a) and 30.1 GHz (D101_LT_A2_F3_C6_01_WG3, Fig. 4b). The full width at half-maximum (FWHM) spectral bandwidth of the 81 GHz single soliton is 4.9 THz, corresponding to a pulse duration of 63 fs. The 30.1 GHz single-soliton state features a bandwidth of 4.0 THz and supports a pulse duration of 71 fs. Various multi-soliton states were also achieved, and we depict an example state with three solitons in Fig. 4e (D101_LT_A2_F2_C4_01_WG3). In each of the ten tested devices with the orientation $\theta = 90°$, solitons were consistently generated. Conversely, none of the attempts to generate solitons were successful in the ten devices oriented at $\theta = 0°$. This demonstrates that altering the crystal orientation to mitigate the Raman effect can be an effective method for generating solitons. The low repetition rate of 30.1 GHz solitons allows the direct detection of the microwave repetition beat note on a fast photodiode. We measured the phase noise of the microwave beat note using an electrical spectrum analyser and found a phase noise level of −86 dBc Hz⁻¹ at an offset frequency of 10 kHz and −114 dBc Hz⁻¹ at an offset frequency of 1 MHz (Fig. 4f), which is higher than earlier measurements using $Si_3N_4$ optical microresonators[44] and in z-cut LiNbO₃ (ref. 42). It is notable that here DKS generation was achieved in an x-cut ferroelectric crystal sample for the first time. This further advances the application of ferroelectric materials for microcomb researchers, given that x-cut samples offer direct access to the largest electro-optic tensor component[1,42].

In summary, we have developed LiTaO₃ PICs that are low loss, exhibit low birefringence and have similar properties to those fabricated from lithium niobate. Crucially, LiTaO₃ is already used commercially in large volumes for wireless filters, thereby providing a path to scalable manufacturing at low cost. Our LTOI PICs achieve similar loss and electro-optical performance to the well-established LNOI technology that has major potential for use in data-centre interconnects[3], long-haul optical communications[4] and quantum photonics[45,46]. The use of low-cost substrates is of central importance for adoption in applications such as data-centre interconnects, in which the die size is large, owing to the requirements of low modulator voltage and the length of travelling wave modulator devices. In our work, we not only establish a smart-cut process for the manufacture of LTOI wafer substrates, but also demonstrate a complete manufacturing process, including the etching of LiTaO₃, the removal of the redeposition of etch products on the waveguide sidewall, and the manufacture of thick metal electrodes for functional electro-optic devices. We also demonstrate key performance metrics, such as low propagation losses of 5.6 dB m⁻¹ and a high electro-optic modulation efficiency of $V_\pi L = 1.9$ V cm at 1,550 nm. Our process is fully wafer-scale and based on deep-ultraviolet photolithography and lays the foundation for the scalable manufacture of high-performance electro-optical PICs that can harness the scale of LTOI wafer fabrication for 5G filters, which is continuing on wafer sizes of 150 mm and 200 mm. Our LTOI platform is particularly promising for applications that can directly exploit the superior properties of the material, such as reduced birefringence. Our platform is capable of processing signals across all optical communications bands (1,260–1,620 nm) in a single PIC, owing to the successful suppression of fundamental mode mixing. It also supports soliton-microcomb generation in the x-cut, whereas for LiNbO₃, soliton-microcomb generation has so far been observed only in the z-cut[1,42], which has compounded the combination of electro-optical and Kerr nonlinearities[47]. Moreover, the low birefringence allows for the ultra-broadband dispersion engineering of LTOI waveguides and for electro-optical frequency comb generation, in which the bandwidth is limited by dispersion[12] and birefringence[48], and octave-spanning bandwidth has not yet been achieved. LTOI is equally promising for the quantum transduction of single microwave photons[21,22], which has recently garnered attention as a way to overcome the thermal bottlenecks of interfacing with superconducting quantum computers[49], because the dielectric loss tangent of LiTaO₃ (ref. 23) is nearly 10-fold lower than that of LiNbO₃ (ref. 24).

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

## Methods

### LTOI wafer fabrication

The thin-film LiTaO$_3$ wafers were fabricated by ion cutting and wafer bonding, starting with ion implantation into a 525-µm-thick bulk LiTaO$_3$ wafer. Commercially available optical-grade and acoustic-grade bulk LiTaO$_3$ wafers were used. Both grades of wafer are congruent compositions and their cost is essentially the same. The difference between acoustic-grade and optical-grade LiTaO$_3$ lies only in whether the material has undergone a chemical reduction process (typically annealed in carbon powder)[31]: acoustic-grade LiTaO$_3$ is reduced but optical-grade LiTaO$_3$ is not. Of the two, the optical-grade LiTaO$_3$ wafers exhibited slightly better crystalline quality, evidenced by a narrower FWHM extracted from the X-ray radiation diffraction rocking curve (Extended Data Fig. 1c,e). Hydrogen ions with an energy of 100 keV and a fluence of $7.0 \times 10^{16}$ cm$^{-2}$ were implanted into a 4-inch x-cut bulk LiTaO$_3$ wafer. An ion-damaged layer was introduced under the surface. Subsequently, the implanted wafer was flipped and bonded to a blank 525-µm-thick high-resistivity silicon carrier wafer covered with 4.7-µm-thick thermal silicon dioxide. A subsequent thermal annealing step (190 °C for 10 h) facilitated the separation of the residual bulk wafer and the exfoliated LiTaO$_3$ thin film. After that, we performed edge removal of the LiTaO$_3$ thin film and chemical mechanical polishing to remove the rough and defect-rich layer of LiTaO$_3$ that was strongly affected by H-ion implantation and thin the LiTaO$_3$ film to the desired thickness of 600 nm. The established process is adaptable to accommodate wafer sizes of 6 or 8 inches.

### LTOI PICs fabrication

We adapted and improved the DLC-based process recently demonstrated for the LNOI platform[7] to fabricate LiTaO$_3$ PICs. The process flow is schematically illustrated in Fig. 1f. First, we deposited a 30-nm-thick layer of Si$_3$N$_4$, a 480-nm-thick layer of DLC and a 60-nm-thick layer of Si$_3$N$_4$ by PECVD as the main hard mask for subsequent IBE. Then we defined the photonic waveguides and components by deep-ultraviolet stepper photolithography and transferred the pattern first into a thin Si$_3$N$_4$ layer by fluorine-based dry etching and subsequently into the DLC hard mask layer by oxygen-based dry etching in a reactive ion etcher. The main etch of the photonic device layer was performed by IBE removing 500 nm of LiTaO$_3$ and leaving a 100-nm-thick continuous LiTaO$_3$ slab across the wafer. After dry etching, it is known for LiNbO$_3$ (ref. 7) that an extra wet etching step with RCA-1 solution is needed because of the non-volatile by-product accumulating on the waveguide sidewall. However, LiTaO$_3$ exhibits not only a higher mass density but also stronger chemical bonds (Extended Data Table 1), resulting in increased mechanical and chemical strength, so a method for removing by-products is not needed for LiTaO$_3$. Here, we removed the LiTaO$_3$ redeposition with a more-alkaline solution of 3:1 KOH(40%):H$_2$O$_2$(30%). After that, we annealed the wafer at 500 °C in an oxygen atmosphere and deposited a 2-µm-thick upper cladding with PECVD based on a hydrogen-free precursor to avoid overtone absorption from optical phonons of the Si–OH stretch vibration around 1.5 µm. The subsequent chip release entailed processes involving dry etching of chip boundaries in SiO$_2$ using fluorine-based chemistry, additional etching of the silicon carrier using the Bosch process, and backside wafer grinding. Apart from the low-loss optical waveguides, high-quality electrodes are also required for the PICs. We develop a DUV stepper-compatible lift-off process with a dielectric sacrificial layer (Extended Data Fig. 5a). After finishing the fabrication of the optical waveguides, we deposited a SiO$_2$ layer with PECVD (step 1) with a thickness that exceeds the desired metal thickness for the lift-off process. Then the pattern was defined by DUV-stepper lithography and transferred to the SiO$_2$ by dry etching. It is critical to undercut the sacrificial layer for the lift-off process, which we performed by dipping the wafer into

buffered hydrofluoric acid (step 2; Extended Data Fig. 5b) for a short time. Metal deposition and the following photoresist lift-off created the metal electrodes (steps 3 and 4). As for the air cladding devices, the SiO$_2$ layer could be removed with another wet etching in buffered hydrofluoric acid (step 5; Extended Data Fig. 5c).

### Photorefractive effect comparison between LiTaO$_3$ and LiNbO$_3$

The measured LTOI (D101_LT_A2_F2_C3_02_WG2) and LNOI (D101_LN_F2_C3_02_WG2) racetrack microresonators have identical waveguide dimensions of 2 µm × 0.5 µm and a slab thickness of 100 nm. Both microresonators are uncladded and have the same free spectral range of 80 GHz. A CW pump laser was Pound–Drever–Hall locked to the cavity resonance and the resonance shift over time was monitored by a wavemeter. During the experiment, the input optical power delivered by the input lensed fibre was maintained at 3 mW for both the LNOI and LTOI samples, which experienced identical fibre–chip coupling losses of 6 dB per facet, resulting in an equivalent power of approximately 0.75 mW in the bus waveguide for each sample. The extinction ratio and loaded linewidth of the resonances were determined using a frequency-comb-calibrated transmission curve measurement, as illustrated in Fig. 2b. Both the LNOI and LTOI samples exhibited an extinction ratio of 50%. The loaded linewidth for the LNOI resonance was measured at 265.4 MHz ($\kappa_0/2\pi$ = 33.4 MHz and $\kappa_{ex}/2\pi$ = 232 MHz), whereas for the LTOI resonance, it was found to be 150.8 MHz ($\kappa_0/2\pi$ = 18.8 MHz and $\kappa_{ex}/2\pi$ = 132 MHz), indicating higher intra-cavity power for the LTOI sample. The resonance frequency shift, denoted as $\Delta f$, over time $t$, induced by photorefractivity, was modelled as a result of the charge-accumulation process: $\Delta f = \Delta f_0(1 - e^{-t/T})$, where $T$ represents the time constant of the charge accumulation process and $\Delta f_0$ is the equilibrium photorefractive-induced frequency shift. The data presented in Extended Data Fig. 3 show a notably smaller time constant, $T$, and equilibrium shift, $\Delta f_0$, for LTOI than for LNOI. This observation is consistent with results obtained from measurements in bulk LiTaO$_3$ crystals, as previously reported[38].

### Electro-optical device characterization

Linear electro-optic tuning measurement was performed in the C-band and O-band using an ECDL. A fibre polarization controller was used to ensure the excitation of the TE mode. The laser frequency was calibrated with a 250 MHz phase modulation by detecting the sidebands around the resonance. A voltage was applied across two of the four electrodes and an oscilloscope was used to record the shift of the resonance to achieve the measurement of the voltage tuning coefficient. For $V_\pi$ measurement, the MZM modulator was driven using a 10 kHz triangular voltage signal while real-time monitoring of the optical transmission signal was done. The extinction ratio was measured to be 15 dB and could be further improved by using directional couplers. For the high-speed electro-optical bandwidth measurement, a pair of high-speed microwave probes was used to deliver the microwave signal to the input port of the transmission line. The output of the transmission line was terminated with a load of 50 Ω. The light was coupled into and collected out of the chip using tapered lensed fibres. The modulated optical signal was pre-amplified and filtered through an erbium-doped fibre amplifier and a bandpass filter, then detected by a 50 GHz photodiode (XPDV2120RA-VF-FP). The $S_{21}$ response (ratio of powers) was measured by a 67 GHz vector network analyser (VNA, R&S ZNA67).

### Soliton and Raman measurement

We used a rapid single-sideband tuning scheme described previously[43] to overcome thermal nonlinearities and initiate solitons at a pump power of 90 mW on-chip using an ECDL and an erbium-doped fibre amplifier for pumping. The optical set-up for soliton generation is depicted in Fig. 4c. The microresonator had a waveguide cross-section of 2 µm × 0.5 µm with a 100-nm-thick slab. Raman measurement was performed using a confocal RM5 Raman microscope. A 532 nm

excitation laser was pumped with a 20× air objective lens and the Raman scattering signal was collected by the same objective. We used a UHTS300 spectrometer with a grating of 1,800 grooves per mm. A half-waveplate was used to change the polarization direction of the excitation laser. The measured Raman spectra of x-cut LNOI and LTOI for the excitation laser with polarization angles of 0°, 30°, 60° and 90° with respect to the $z$ axis are depicted in Extended Data Fig. 4a,b. Both samples have several strong vibration phonon branches with large linewidths. Typical Raman peaks are labelled. The strongest peak at 517 cm$^{-1}$ corresponds to the silicon substrate. The Raman intensity becomes lower when the polarization of incident light transitions from being parallel to the $z$ axis to being parallel to the $y$ axis. Remarkably, this reduction is more pronounced in LTOI than in LNOI (Extended Data Fig. 4c). We therefore used racetrack microresonators with the straight waveguide section oriented along the $z$ axis to minimize the Raman interference. This configuration ensured that the TE mode predominantly aligned with the non-polar $y$ axis, as depicted in the schematic chip diagram in Fig. 4c,d.

## Material properties comparison between LiTaO₃ and LiNbO₃

LTOI not only has the advantage of higher production volume but also has similar or even better performance than LNOI, owing to the inherent properties of LiTaO₃. A summary of a comparison of the material properties of LiTaO₃ and LiNbO₃ is shown in Extended Data Table 1. All the properties listed correspond to the congruent compositions, which are both more readily available and more widely used in various fields than their stoichiometric counterparts.

## Data availability

The data used to produce the plots in this paper are available at Zenodo at https://doi.org/10.5281/zenodo.10215427 (ref. 50).

## Code availability

The code used to produce the plots in this paper is available at Zenodo at https://doi.org/10.5281/zenodo.10215427 (ref. 50).

50. Wang, C., & Kippenberg, T. J. Lithium tantalate photonic integrated circuits for volume manufacturing. *Zenodo* https://doi.org/10.5281/zenodo.10215426 (2023).

**Acknowledgements** The samples were fabricated in the EPFL Center of MicroNanoTechnology (CMi) and the Institute of Physics (IPHYS) clean-room. The LTOI wafers were fabricated in Shanghai Novel Si Integration Technology (NSIT) and the SIMIT-CAS. This work has received funding from the National Natural Science Foundation of China (62293521). T.J.K. acknowledges funding from the European Research Council grant 835329 (ExCOM-cCEO) and from the EU Horizon Europe research and innovation program through grant 101113260 (HDLN). J.R. acknowledges funding from the SNSF through Ambizione Fellowship 201923. C.W. acknowledges financial support from the China Scholarship Council (202104910464). X.O. acknowledges the National Key R&D Program (2022YFA-1404601) from the Ministry of Science and Technology of China.

**Author contributions** C.W. and Y.C. fabricated the LTOI wafers with technical support from K.H. and X.W. C.W. and Z.L. fabricated the devices. J.R., Z.L. and C.W. performed numerical simulations and designed the devices. C.W. and Z.L. characterized the samples. C.W., Z.L., G.L., J.Z., M.C., X.J., A.D., T.B. and W.K. performed experiments. J.R., C.W, G.L. and Z.L. analysed the data, prepared the figures and wrote the manuscript with input from all authors. T.J.K. and X.O. supervised the project.

**Funding** Open access funding provided by EPFL Lausanne.

**Competing interests** T.J.K. is a co-founder and shareholder of Luxtelligence SA, a foundry commercializing LiNbO₃ PICs, as well as DEEPLIGHT SA, a start-up commercializing PIC-based frequency-agile low-noise lasers.

**Additional information**
**Correspondence and requests for materials** should be addressed to Xin Ou or Tobias J. Kippenberg.

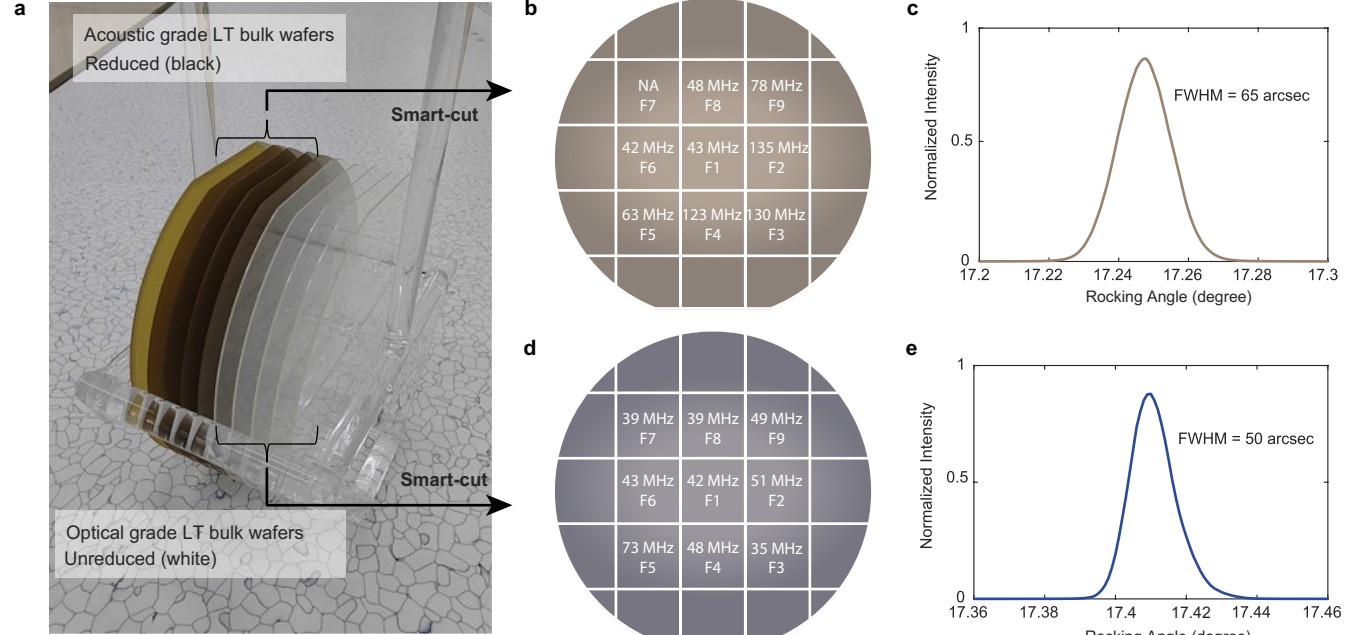

**Extended Data Fig. 1 | Optical loss of the acoustic and optical grade LTOI integrated photonic devices.** (a) Photo of the bulk acoustic and optical grade LiTaO₃ wafers. The black and yellow wafers are reductive and original acoustic grade LiTaO₃ bulk wafers, respectively. The white wafers are optical grade. (b) Wafer map of the optical loss of LiTaO₃ microresonators fabricated on acoustic grade LTOI wafers. The resonators are 2 μm wide racetracks with 80 GHz free spectral range. Figure shows the most probable values of the intrinsic loss rate $\kappa_0/2\pi$ for a resonator from every field. The test chip from one field on the acoustic grade wafer was damaged before the measurement (shown as NA). (c) XRD rocking curve measured on the LiTaO₃ (110) Bragg peak of the acoustic grade LTOI wafer after bonding and polishing. (d,e) Same as (b,c) but for optical grade LTOI wafer.

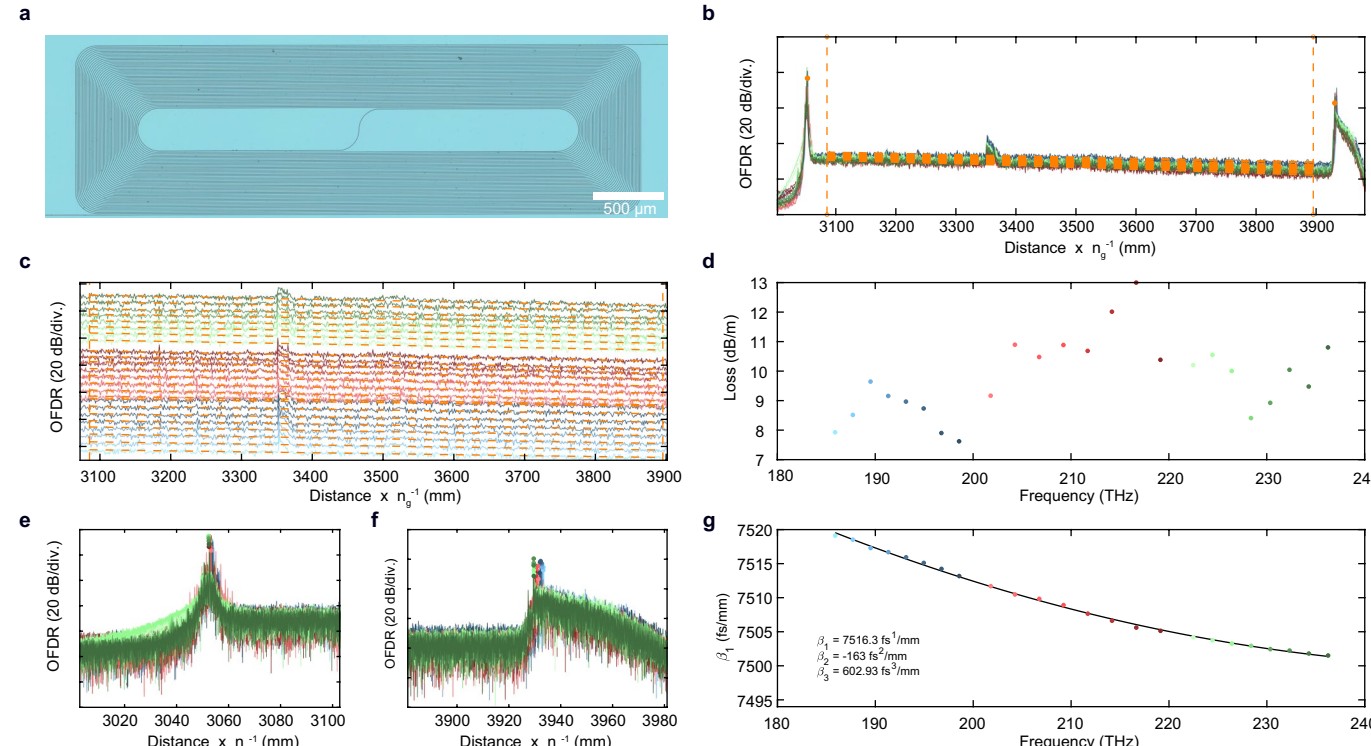

**Extended Data Fig. 2 | Linear loss and dispersion measurement.** (a) Optical micrograph of LTOI spiral (D101_LT_A2_F2_C7_Spiral5) with a length of 39 cm and a waveguide cross-section 1.75 $\mu$m × 0.6 $\mu$m. The rectangular footprint is 5 × 2 mm². (b) Segmented Fourier transform of the optical frequency domain reflectometry (OFDR) signal showing the strength of the coherent optical backreflection as a function of the optical length. Colors encode the central wavelength of the segments and correspond to illustrations in panel (d). An optical distance of 88 cm can be identified according to the reflection peaks of the front and back facets of the waveguide spiral chip. A minor fabrication defect is found at an optical distance of 3.36 m. (c) Same as panel (b) but individual traces offset by 3 dB to highlight the linear fit for extraction of the optical propagation loss. (d) Optical propagation losses extracted from the fits in panel (c) showing a propagation loss of around 8 dB/m in the optical C-and L-bands and around 10 dB/m in the O-band. (e,f) Same as panel (b) but highlighting the regions around the front and back facet reflections of the chip. The colored markers point to the extracted position of the facet and are used to infer the dispersion of the spiral. (f) Frequency-dependent optical group velocity $v_g = \beta_1^{-1}c = 2.25$ of the waveguide spiral and fit of the anomalous dispersion profile of $\beta_2 = -163$ fs mm⁻². The dispersion contribution of the 300 $\mu$m long tapers is neglected.

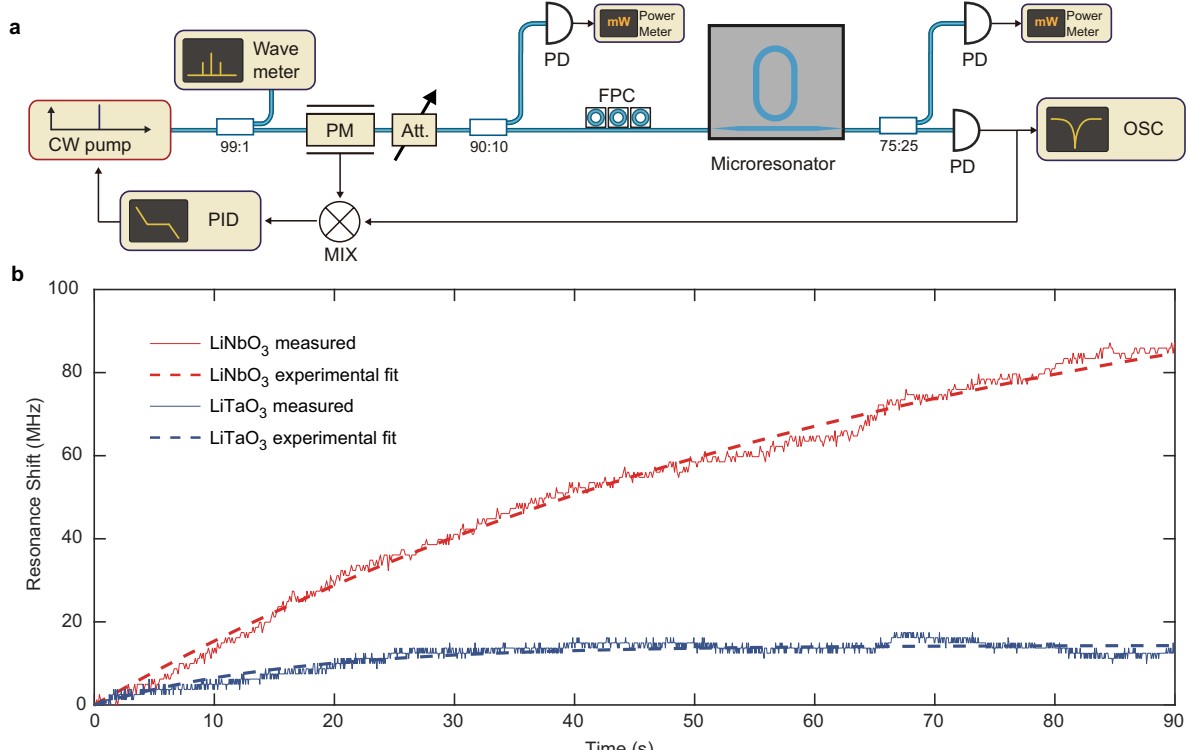

**Extended Data Fig. 3 | Photorefractive effect comparison between LiTaO₃ and LiNbO₃.** (a) Schematic of resonance drifting measurement setup. A pump laser is Pound-Drever-Hall (PDH) locked to the resonance microresonators and the wavelength is monitored by a wavemeter. (b) Photorefractive-induced resonance frequency shift in LNOI (red) and LTOI (blue) microresonators.

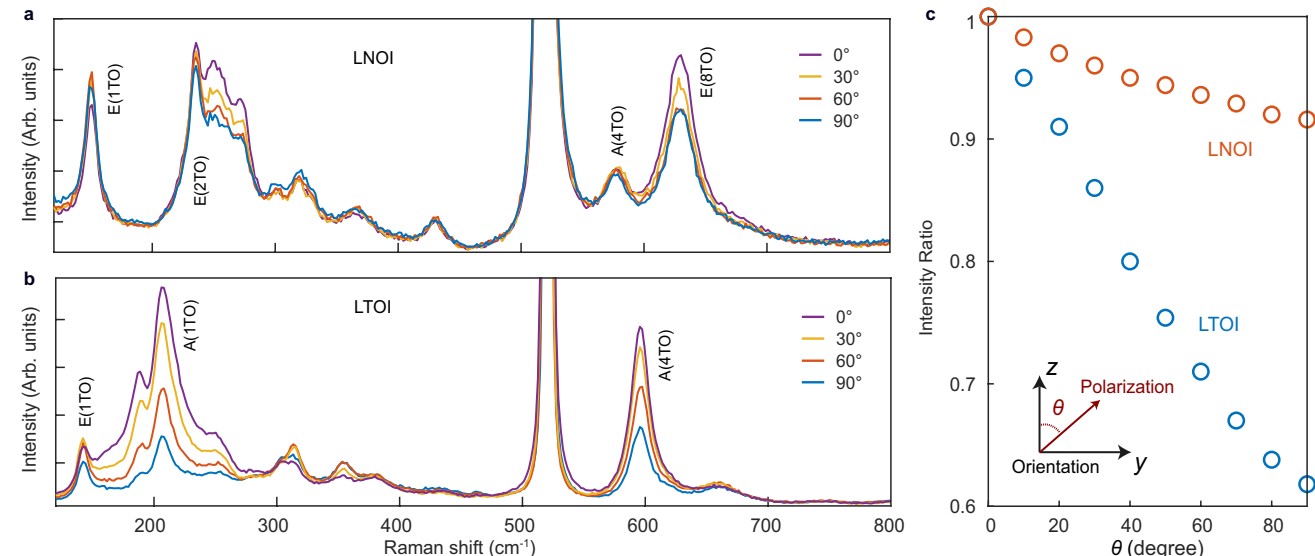

**Extended Data Fig. 4 | Raman intensity comparison between LNOI and LTOI.** Raman spectra of (a) x-cut LNOI and (b) LTOI for the excitation laser with polarization angles 0°, 30°, 60°, 90° with respects to the z-axis. (c) Variation of the Raman intensity ratio of LNOI and LTOI for the excitation laser with different polarization angles. The Raman intensity is obtained by integrating all peaks that correspond to LiNbO$_3$ or LiTaO$_3$ and is normalized. Inset illustrates the angles between the orientation and the polarization of the excitation laser.

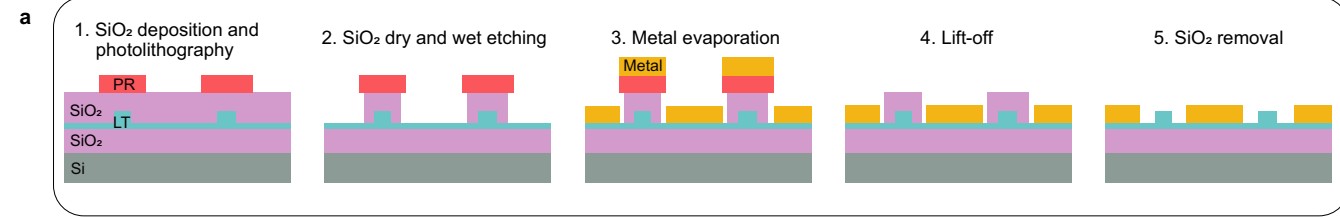

**a**

1. SiO₂ deposition and photolithography
2. SiO₂ dry and wet etching
3. Metal evaporation
4. Lift-off
5. SiO₂ removal

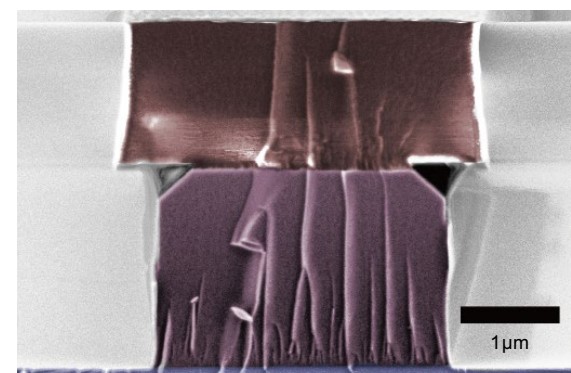

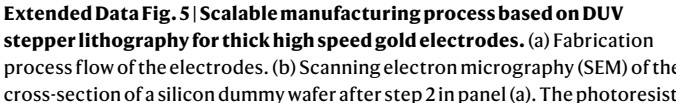

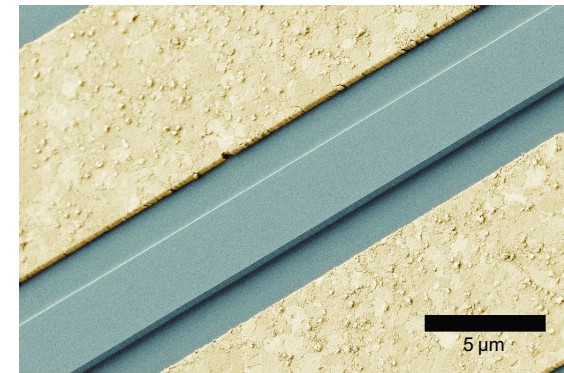

**b**

1 µm

**c**

5 µm

**Extended Data Fig. 5 | Scalable manufacturing process based on DUV stepper lithography for thick high speed gold electrodes.** (a) Fabrication process flow of the electrodes. (b) Scanning electron micrography (SEM) of the cross-section of a silicon dummy wafer after step 2 in panel (a). The photoresist is colored in red, silicon dioxide in pink, and silicon in blue. (c) SEM of fabricated electrodes and LiTaO₃ waveguide. The gold is colored in yellow and LiTaO₃ in light blue.

**Extended Data Table 1 | Comparison of the material properties of LiTaO₃ and LiNbO₃**

| Properties, congruent | LiTaO$_3$ | LiNbO$_3$ |
|---|---|---|
| Lattice constant, Å | a = 5.154, c = 13.783 | a = 5.148, c = 13.868 |
| Density, $\mathrm{g/cm^3}$ | 7.45 | 4.64 |
| Curie point, °C | 610 | 1157 |
| Band gap [18,19], eV | 3.93 | 3.78 |
| Dielectric Constant [48,49] @ $100\,\mathrm{kHz}$ | $\epsilon_{11,22} = 54, \epsilon_{33} = 43$ | $\epsilon_{11,22} = 38, \epsilon_{33} = 28$ |
| Refractive index @ $1550\,\mathrm{nm}$ | $n_o = 2.119, n_e = 2.123$ | $n_o = 2.21, n_e = 2.14$ |
| Birefringence | $\Delta n = 0.004$ | $\Delta n = -0.07$ |
| Electro-optic coefficient [1] @ $1550\,\mathrm{nm}$, pm/V | $r_{33} = 30.5, r_{51} = 20$ | $r_{33} = 30.9, r_{51} = 32.6$ |
| Second nonlinear coefficient @ $1064\,\mathrm{nm}$, pm/V | $d_{33} = -21$ | $d_{33} = -27$ |
| Optical damage threshold @ $1064\,\mathrm{nm}$, $\mathrm{MW/cm^2}$ | 240 | 120 |
| Coercive electrical field @ $\mathrm{kV/mm}$ | 21 | 21 |

The lattice constant, density, Curie point, bandgap, dielectric constant, refractive index, birefringence, electro-optic coefficient, optical damage threshold and coercive electrical field are compared.