## [Peer Review File · Nature]

Manuscript Title: Lithium tantalate photonic integrated circuits for volume manufacturing

Reviewer Comments & Author Rebuttals

Reviewer Reports on the Initial Version:

Referees' comments:

Referee #1 (Remarks to the Author):

The authors establish a smart-cut process for the manufacturing of LTOI wafer substrates, demonstrate a complete manufacturing process including the etching of LiTaO₃, and developed lithium tantalate photonic integrated circuits that are low loss, exhibit low birefringence and have comparable properties to Lithium Niobate. There are three major components to this manuscript,

1. The fabrication process for LTOI wafers and optical waveguides,
2. A tunable high Q microresonator and push-pull MZI
3. The dissipative Kerr soliton (DKS) generation

These results will be of interest to the LTOI community, especially for the high quality LTOI wafer. However, the performance (loss, modulation efficiency) of the optical waveguide, microresonator, and the push-pull MZI are commonplace, comparing to the existing similar LNOI or SiN devices. The advantages of LTOI mentioned in the manuscript are not well reflected, according to the performance of these devices.

1. The authors claimed that “they overcome the challenge (high cost per wafer, and limited wafer size) and demonstrate a photonic platform that satisfies the dichotomy of allowing scalable manufacturing at low cost -based on today’s existing infrastructure—while at the same time exhibiting equal, and superior properties to those of Lithium Niobate” . But There is no direct research comparison on the cost and wafer size between LNOI and LTOI in the manuscript. Actually, LNOI has been commercialized and 8-inch lithium niobate wafers have also been implemented. And the LTOI in the text appears to be 4 inch. The advantages of the preparation process are also not significant, comparing to LNOI fabrication as ref 7 in the manuscript.

2. Comparing the MZI performance with the LNOI modulators, the modulation efficiency is not prominent. And there seem to be no bandwidth parameters of the LTOI MZI. The low propagation loss and high electrooptical efficiency of LTOI claimed in the manuscript are not clearly reflected.

3. For the LTOI Soliton microcomb, What are the advantages of LTOI compared to other materials (SiN, LNOI ...)to generate soliton combs, especially for LNOI? The authors may need to study why the soliton microcomb can not be generated in X-cut LNOI, and what is the difference for the soliton microcomb generation between X-cut LTOI and LNOI, and have an overall comparison on those in Z- or X- cut crystal thin films?

Referee #2 (Remarks to the Author):

In this manuscript, comprehensive results on the fabrication of various integrated-optical photonic devices (micro-resonators, electro-optical modulators, soliton micro-combs) in the new thin-film material lithium tantalate (LTOI) are presented for the first time in an overview. Compared to the thin-film material LNOI based on lithium niobate, which has been intensively investigated for several years and is basically very similar in terms of optical properties, LTOI exhibits small but important differences. These include higher transparency in the near UV range, higher optical damage thresholds and significantly lower birefringence, which is important for individual optical functions. Previous work dealing mainly with the use of LNOI for photonic devices is appropriately referenced, and the references given clearly support the understanding of the subject. Abstract, the conclusions and summary of the paper are clear and sufficiently informative.

The demonstrated results on photonic integration using the new LTOI platform developed by the authors are very novel, of very high scientific quality and very promising for the development of photonic integrated circuits (PICs); the here presented first results demonstrate already a lot of challenging high-level experimental work. The work submitted here, with a focus also on demonstrating the potential for mass (i.e. wafer scale) production of PICs, has in principle the potential to be highly interesting and valuable to a wider scientific community - beyond the field of photonics - and to lay the foundation for further research efforts on this new optical material platform. The scientific impact of this work is expected to be high. Overall, I would recommend the manuscript for publication in Nature, provided the authors consider the further improvements suggested below.

Some points that should be revised/improved are:

- both abstract and top of page 2: The sentence “We show that LiTaO₃ [...] can be etched with high precision [...] using DUV lithography [...] and alkaline wet etching.” is misleading because in this work LiTaO₃ is dry etched using IBE. Please clarify this part.
- in the abstract it is claimed that “LiTaO₃ [...] exhibit higher photorefractive damage threshold.” However, this claim is not supported by any reference. Please either skip this statement or justify it by proving adequate literature sources.
- on top of page 2 it reads “The stronger chemical bonds induce a higher electron density in LiTaO₃”. The meaning of this sentence is not clear for me and seems to be physically incorrect. The number of valence (binding) electrons in LiNbO₃ and LiTaO₃ is identical, although bonds in LiTaO₃ of course are stronger. In total Ta ions have more electrons than Nb ones, but as these are strongly bonded to the cores they do not play a significant role for the bonding strengths or polarizability/refractive index (indeed indices for LiNbO₃ are slightly higher), so an effect of “higher electron density” on material properties is not clear here. The 2nd part of the sentence says “exhibit not only higher density”; here it is unclear which density (electron or mass?) the authors mean.
- the given band gap values E_g on top of page 2 (although measured in [18]) are not consistent with most other data from literature
- the composition of the used optical-grade LiTaO₃ is missing – I assume it is congruently melting

one. However, in the supplementary section on page 1 the sentence “the optical grade LiTaO₃, which has fewer crystal defects and is closer to the stoichiometric LiTaO₃ crystal” is irritating in this context. Please clarify

- page 2: Curie temperature for LiTaO₃: I suggest to write “(610°C – 700°C, depending on the Li/Ta ratio)”

- Reference [3] by C. Wang et al. appears again as number [11]

- the authors highlight several times that they “propose a new solution to remove LiTaO₃ redeposition”, but any details on how this is performed are missing on page 3 (it now reads “with an additional wet etch step”). Please add a detailed description of the method that allows to use this method also by others.

- The different content of Fig. 1 seems to be not well organized. In my opinion the (tiny) crystal structure can be skipped. The rest of this figure should be organized in a more logical order, i.e. starting with smart cut (f), showing wafer pictures (g,h) and properties (l,j), followed by waveguide fabrication (e) and showing examples of fabricated structures (c,d,b).

- top of text on page 4: should be Figure 2c (not 3c)

- In Fig. 2 part (e), showing a qualitative index ellipsoid (here for negative birefringence, i.e. for LiNbO₃), could be skipped in my opinion as it provides not much information for the reader. The same for part (f), without further details there seems to be no useful information for the reader. Furthermore, this figure part is never referenced in the text.

- 2nd paragraph of page 4: better write “magnitude of optical birefringence” (because +0.004 is larger than -0.07)

- Figure 3(f) seems to be not referenced in the text

- Figure 4(e) seems to be not referenced in the text

- the first section of the supplementary information is of low quality overall and the necessity to provide all the data in Table 1 is not obvious. As far as optical-grade materials are concerned, the production volume of LiNbO₃ might be far higher than that of LiTaO₃, also costs of such LiNbO₃ wafers are much lower. LiTaO₃ not only has some superior properties than LiNbO₃, but also has the disadvantage that its nonlinear coefficient is smaller than that for LiNbO₃, resulting in much lower nonlinear conversion efficiencies. Anyway the given d_{33} value for lithium tantalate seems to be wrong; in literature it is typically described as negative with a size of -21 pm/V. In my opinion it makes no sense to classify optical damage thresholds as “high” or “low” without giving numbers - as far as I know, this has not even been studied quantitatively for LNOI (or LiTaO₃), it is also strongly wavelength dependent. The term “poled availability” makes little sense in terms of content; what the authors may mean is the ability to achieve (quasi-) phase matching by periodic poling. Birefringence is defined as $\Delta n = n_e - n_o$, i.e., it is negative for LiNbO₃ (please correct this in table 1). The lower transparency limit for LiNbO₃ is more like ~350nm instead of the value of 400nm given now. It should be added to Table 1 that all parameters apply to the congruently melting compositions. The band-gap values given in table 1 (from [18], obtained for an absorption coefficient of 300cm⁻¹ for not further specified crystals/compositions) are not consistent with most other literature data and are in disagreement with the given lower transparency ranges/wavelengths. A good overview for LN can be found in: Results in Physics 39, 105736 (2022).

- In the section on linear losses of the supplementary material the fabricated spiral is claimed first to have a length of 39cm, while later in the text it has 90cm length. Please check this.

**Resubmission of a revised version of
Nature Manuscript Number 2023-06-11067**

“Lithium tantalate photonic integrated circuits for high volume manufacturing”

Dear Reviewers,

We are grateful that two reviewers have seen our manuscript. After reading the comments made by the referees thoroughly, we would like to thank them for the detailed review and suggestions to improve the manuscript. We appreciate that the reviewers gave a positive evaluation such as:

“These results will be of interest to the LTOI community, especially for the high quality LTOI wafer.”
(Reviewer #1)

“The demonstrated results on photonic integration using the new LTOI platform developed by the authors are very novel, of very high scientific quality and very promising for the development of photonic integrated circuits (PICs); the here presented first results demonstrate already a lot of challenging high-level experimental work. The work submitted here, with a focus also on demonstrating the potential for mass (i.e. wafer scale) production of PICs, has in principle the potential to be highly interesting and valuable to a wider scientific community - beyond the field of photonics - and to lay the foundation for further research efforts on this new optical material platform. The scientific impact of this work is expected to be high. Overall, I would recommend the manuscript for publication in Nature, provided the authors consider the further improvements suggested below.” (Reviewer #2)

Before responding individually to each reviewer, we would like to first state an **Executive Response** to all reviewers to highlight the main novelty of our work:

We emphasize that our work's **key novelty** and **scientific significance** is the first demonstration and development of photonic integrated circuits based on lithium tantalate-on-insulator (LTOI) wafers. We demonstrate low optical loss and in tightly confining Lithium Tantalate strip waveguides and demonstrate their application for electro-optic modulators. Similar to LiNbO_3 , LiTaO_3 is difficult to etch and we developed a novel recipe that enables the removal of re-deposition and enables record-low loss waveguides with 5.6 dB/m loss. In addition, we can harness the 2nd-order Pockels and 3rd Kerr nonlinearity and demonstrate fast electro-optical modulation and soliton microcomb generation in X-cut LTOI. This implies it is possible to combine soliton microcomb generation and high-speed electro-optical modulation in a single monolithic photonic platform. This is noteworthy as prior work on LNOI could achieve soliton generation, but only in z-cut and not in X-cut LNOI, which compounds the seamless integration of the soliton microcomb generators and electro-optical modulators, due to the out-of-plane electrical field that is required. we demonstrate experimentally, that LTOI integrated photonics exhibit significantly lower birefringence, allowing devices that operate mode-crossing-free over all currently employed telecommunication bands (from O- to the L-band).

Our results are not only of scientific importance but also crucial to the future adoption of electro-optical photonic integrated circuits. Today Lithium Niobate wafers are costly and not available in high volumes – as is the case for silicon on insulator. The latter can significantly compound the most interesting applications of this technology in high volumes for data center HPC computing interconnects used in ML and AI. For industrial adoption of the technology in such high volume ‘pluggable optics’, the cost and availability of large wafer sizes are critical. LiTaO_3 , as it is a material has been adopted for 5/6G wireless filters due to the existence of a zero-temperature coefficient for acoustic waves. The high-volume manufacturing is critical for future industrial adoption – e.g., for future interconnects for ML and AI computing centers which are both price-sensitive as well as supply chain sensitive. In contrast, LiNbO_3 does not exhibit such properties and LNOI remains a niche material only used for integrated photonics and unable to leverage the technological and manufacturing gravity of the electronics industry that has been the foundation of the success of the SOI optics platform.

In our revised manuscript, we added measurements of an electro-optic Mach-Zehnder LT modulator (MZM) with a 3 dB electro-optical bandwidth of 40 GHz and a half-wave electro-optic modulation efficiency of as low as $V_{\pi} \cdot L = 1.6 \text{ V} \cdot \text{cm}$. We added additional measurements and calculations regarding the Raman nonlinearity, comparing spectra at different crystal orientations for LiNbO_3 and LiTaO_3 , which explain the soliton microcombs in an X-Cut wafer crystal.

In addition, following the request of the editor, we have modified the title to “**Lithium tantalate photonic integrated circuits for high volume manufacturing**”. We appreciate the editor’s suggestion as it aptly reflects our comprehensive exploration of the performance of LiTaO_3 in integrated photonics, including electro-optical aspects, third-order nonlinearity, high-volume manufacturing, and broad bandwidth capabilities. We added more details of the fabrication method, the material composition, and the design and performance characterizations of the device.

We have addressed all the concerns and suggestions raised by the Referees and formulated our point-by-point response below. Below, we provide a point-by-point response to the reviewers’ comments and questions. The **original review report is printed in black, our responses in blue, and the action taken in red**. In the revised manuscript, according to the reviewers’ request, we have changed the title and added new data. We believe that these comments contribute to the paper’s readability and clarity.

Referee #1 (Remarks to the Author):

First of all, we would like to thank the referee for his comprehensive review, which helped us to improve and clarify our manuscript. Below, we provide step-by-step comments on the concerns expressed and indicate the corresponding changes made to our work.

“The authors establish a smart-cut process for the manufacturing of LTOI wafer substrates, demonstrate a complete manufacturing process including the etching of LiTaO_3 , and developed lithium tantalate photonic integrated circuits that are low loss, exhibit low birefringence and have comparable properties to Lithium Niobate. There are three major components to this manuscript,

1. The fabrication process for LTOI wafers and optical waveguides,
2. A tunable high Q microresonator and push-pull MZI.
3. The dissipative Kerr soliton (DKS) generation.

These results will be of interest to the LTOI community, especially for the high quality LTOI wafer.”

We thank the reviewer for the positive assessment. We would like to clarify that in addition to the points mentioned above, our work also demonstrated record low-loss in a high yield, wafer scale manner with losses of 5.6 dB/m using a novel processing method (that is distinct from LNOI). In addition, we would like to emphasize that we also prove the lower birefringence experimentally in a microresonator that operates over all telecommunication bands (cf. Figure 2). Last, while not a scientific achievement, our work is also technologically important in identifying a material that has not only superior optical properties but crucially also a path towards volume manufacturing in other applications. As outlined above LTOI is an ideal material for 5/6G filter applications that also require an insulation layer of oxide. As such the low costs and large wafer size that LTOI wafers will develop in the coming years, with rising adoption in wireless 5/6G will allow to significantly reduce the wafer price and as such open the path for electro-optical PICs to entire cost-sensitive large volume applications, such as pluggable optics for data center interconnects, as well as disaggregated computing architectures for ML and AI. In addition, the lower energy, and the use of Hydrogen instead of Helium, of implants make LTOI wafers inherently less costly than LNOI.

“However, the performance (loss, modulation efficiency) of the optical waveguide, microresonator, and the push-pull MZI are commonplace, compared to the existing similar LNOI or SiN devices. The advantages of LTOI mentioned in the manuscript are not well reflected, according to the performance of these devices.”

We respectfully disagree with the criticisms of the device's performance and explain why.

First, concerning the **optical loss**: The comparison to Si_3N_4 is not proper as the Si_3N_4 material is amorphous and lacks a Pockels coefficient. However, even when compared to commercial Si_3N_4 the optical losses in our work are lower. Only more ‘exotic’ manufacturing techniques (i.e., electron beam lithography or reflow or low-confinement waveguide) allow higher Q at the level of 1.4 dB/m. The table below showcases a comparison between our work and the most advanced developments within LNOI photonics from the past decade. While the loss values of the LNOI microresonator reported in Ref [1] slightly surpass ours, their study lacks statistical analysis and isn't available for wafer-level fabrication. The works in Ref [3] represent a commercial and academic wafer-level fabrication performed with DUV lithography and are the closest comparison to our process in the literature. However, only a loss of 20 dB/m was achieved in this report. Our own work on LNOI [4] achieves a slightly better loss than [3] on the same level as the present report, our work provides a more unique performance of LTOI and also demonstrates highly efficient electro-optical devices and soliton devices. Notable that the LNOI work selected for comparison represents the pinnacle of advancements in LNOI photonics over the past decade. Yet, we would like to emphasize that LTOI has a slew of unique features both from a scientific perspective (low birefringence, higher damage threshold) and an application-oriented perspective

(existing volume manufacturing and adoption in the electronics industry, that set LTOI apart from LNOI. Given this context, we believe it is false to categorize our LTOI devices' loss performance as commonplace.

Table R 1. Comparison of Q-factor, linear optical loss, EO efficiency, and wafer-level fabrication ability

Ref	Q-factors	Linear optical loss	Statistical analysis	V * L product EO efficiency	Wafer-level fabrication	Platform
This work	7.2×10^6	0.056 dB/cm	Yes	1.9 V cm	Yes	LTOI
[1]	$\approx 10^7$	0.027 dB/cm	No	No data	No	LNOI
[2]	No data	No data	No	2.2 V cm	No	LNOI
[3]	1.8×10^6	0.27 dB/cm	Yes	No data	Yes	LNOI
[4]	$\approx 10^7$	0.04 dB/cm	Yes	3.3 V cm	Yes	LNOI

Concerning the modulation efficiency, in our revised manuscript, we added the MZI device with an electro-optical modulation efficiency of 1.9 V cm. We also list the modulation efficiency inside the table, from which we can see that our work reaches a better modulation efficiency. For a detailed discussion about the modulation efficiency, please see our following reply on MZI performance.

Showing some of the advantages of the LTOI platform, we point out that the low birefringence of the LTOI platform solves long-standing issues with LNOI. LTOI is capable of processing signals across all optical communications bands (1260-1620 nm) in a single LTOI microresonator (cf. Figure 2 e-i). Moreover, the low birefringence allows for ultra-broadband dispersion engineering of LTOI waveguides and also for electro-optical frequency comb generation, where the state-of-the-art bandwidth in LNOI is limited by dispersion [5] and birefringence [6]. We believe that deep etching, flat dispersion, and mode-free coupling can be achieved simultaneously in LTOI, which holds great promise for an octave-spanning EO comb generation. Figure R1(c) shows the simulated dispersion in a deep-etched LTOI microresonator with the dimension of 700 nm x 2500 nm, a flat dispersion can be achieved. Figure R1(c) further reveals that it is possible to achieve an octave-spanning EO comb even at low drive voltages (Simulation result, not shown in the current manuscript).

Figure R1: (a), (b) Simulation of the optical mode and electrical field (bottom) at the LT waveguide with a dimension of 700 nm x 2500 nm. (c) Comparison between LTOI and LNOI for dispersion engineering and tailored EO comb phase-matching condition. (d) Simulated EO comb generation for two different modulation depths in the LTOI platform.

[1] M. Zhang et al., "Monolithic ultra-high-Q lithium niobate microring resonator," *Optica*, 2017
[2] Wang, Cheng, et al. "Integrated lithium niobate electro-optic modulators operating at CMOS-compatible voltages." *Nature* 562.7725 (2018): 101-104.
[3] K. Luke et al., "Wafer-scale low-loss lithium niobate photonic integrated circuits," *Optics Express*, 2020

[4] Li, Zihan, et al. "High density lithium niobate photonic integrated circuits." *Nature Communications* 14.1 (2023): 4856.

[5] Zhang, M. et al. Broadband electro-optic frequency comb generation in a lithium niobate microring resonator. *Nature* 568, 373–377 (2019).

[6] Hu, Y. et al. High-efficiency and broadband on-chip electro-optic frequency comb generators. *Nature Photonics* 16, 679–685 (2022).

Last, as we have outlined in the overall reply it is not only about the superior material properties but crucially, also about identifying a material that allows electro-optical photonic integrated circuits to actually be deployed in large volume applications. By demonstrating and developing LTOI for integrated photonics, we use a material that has a volume market *that already exists today* (i.e. 5G/6G), which is essential to large-scale adoption. We emphasize that our results also have for the first time established a comprehensive process flow that includes efficient wafer production and DUV-compatible lithography and etching. We demonstrate ultra-low optical propagation loss, electro-optical tuning, switching via the Pockels effect, and soliton microcomb generation via the optical Kerr effect of LiTaO₃.

In conclusion, we respectfully disagree with the notion that the advantages of LTOI are not well-reflected. On the contrary, our results clearly reflect the advantages of LTOI over LNOI across many aspects, including mass production capabilities, lower birefringence effect, and Raman suppression. To provide a clear overview of the advancements achieved in this work, we summarize them as the following:

- We demonstrate a smart-cut process for producing high-quality LTOI using **H⁺ ion implantation with implantation energy significantly lower** (ca. 100 keV) than that used for LNOI wafer preparation (ca. 300 keV). This process has substantial implications for reducing production costs and enhancing the efficiency of LTOI wafer manufacturing compared to LNOI wafers.
- We demonstrate for the first-time propagation losses **on par** with LNOI and develop for the first-time etching techniques for low-loss LTOI integrated waveguides.
- We demonstrate for the first time LTOI MZM with a high electro-optical modulation efficiency (1.9 V cm) and a large bandwidth (40 GHz).
- Figure 3e in our manuscript reflects that the LTOI platform can process signals across all optical communications bands (**1260-1620 nm**) in a single photonic integrated circuit without mode mixing, which is not achievable with LNOI due to the higher birefringence. This could enable future broadband EO combs.
- We demonstrate the advantage of LTOI in suppressing the Raman effect, enabling us to achieve **soliton generation for the first time in an x-cut ferroelectric platform**. This enables microcombs with efficient electro-optical modulation (in contrast to Z-cut which requires out-of-plane electrodes).

We added the citation to the paper presenting LTOI material properties: (<https://www.mdpi.com/2073-4352/13/8/1233>)

“Lithium tantalate (LiTaO₃, or LT) is widely used in filters [1], pyroelectric detectors, holographic memory devices [2], and other devices, due to its excellent piezoelectric, pyroelectric, and nonlinear optical properties. Some properties of LT are better than those of lithium niobate (LiNbO₃, or LN) [3], which is isomorphic to lithium tantalate. First, the photorefractive resistance of LT is twice that of LN [4]. Second, LT has a smaller temperature frequency coefficient (TCF) than LN, so it is an ideal material for manufacturing high-stability and broadband filters [5]. Finally, the thermal stability of LT, which is only 18 ppm in the range of 20–80 °C, is better than that of LN. Moreover, the optical damage threshold for LT at 514.5 nm is 1500 W/cm², which is approximately 37.5 times the optical damage threshold for LN, which is only 40 W/cm² [6]. Therefore, LTOI is superior to

LNOI for many application cases from optical modulation and switching as well as nonlinear optical frequency conversion.

Action taken: We added additional citations to show the advantages of the LTOI material. We added new data on MZM characterization (cf. Fig. 3). We compared the Raman intensity between LNOI and LTOI (cf. Fig. 4 and Extended Data Fig. 4).

“1. The authors claimed that “they overcome the challenge (high cost per wafer, and limited wafer size) and demonstrate a photonic platform that satisfies the dichotomy of allowing scalable manufacturing at low cost -based on today’s existing infrastructure–while at the same time exhibiting equal, and superior properties to those of Lithium Niobate”. But There is no direct research comparison on the cost and wafer size between LNOI and LTOI in the manuscript.”

We thank the reviewer for this comment. Comparing the manufacturing costs of LNOI and LTOI, the following two primary factors should be considered:

- **Fabrication Process:** In our manuscript, we described the fabrication process of the ion-implantation for LTOI as follows “We implanted the hydrogen ion into an X-Cut bulk LiTaO₃ wafer with an energy of 100 keV and a dose of $3.2 \times 10^{16}/\text{cm}^2$ ”. In contrast, the LNOI smart-cut process, as detailed by Jia, Y. *et al* [1] and widely applied for wafer production today, requires significantly higher energy, typically ranging from 200-400 keV, and involves helium ions. These differences in ion type and energy have significant implications for production costs and efficiency. Hydrogen ion implantation aligns more closely with the processes used in today's high-volume commercial production of SOI wafers and is hence preferred. Regarding implantation energy, commercial ion implanters usually have a maximum working energy of 200 keV, necessitating upgrades, such as additional acceleration tubes, to reach an energy typically greater than 225 keV. This implies that, in contrast to LiNbO₃, existing machinery can be used for LiTaO₃ thin film fabrication.
- **Production Volume:** The production volume of LTOI wafers exceeds that of LNOI by several orders of magnitude driven by its widespread use in acoustic filters for 5G communications due to its superior temperature stability and low loss performance. In contrast, LNOI lacks a comparable penetration into consumer electronics markets supporting the build-out of large-scale wafer manufacturing lines. According to the *published production status of Soitec*, a global leader in semiconductor materials, the production volume of 150 mm LTOI (referred to as Piezo-OI in their report) is projected to reach 750k wafers per year by 2024 [2]. In contrast, the production volume of LNOI wafers is less than 10k wafers per year, estimated according to the revenue of the two main suppliers - NanoLN and NSIT from China.

Due to the aforementioned factors, LTOI has entered the consumer electronics market ahead of LNOI, which is characterized by significantly higher production volumes compared to the optics industry. Since 2017, LTOI has gained widespread adoption by major players such as **Murata** (the largest surface acoustic wave (SAW) filter company) [3], **Qualcomm** [4], **Qorvo**, and others. These companies have embraced LTOI to fabricate high-performance SAW radiofrequency filters, recognizing the advantages they offer in terms of better performance and reliability. Currently, 150 mm LTOI has become a standard choice for mainstream RF filter companies in the production of high-performance SAW filters. As indicated in Figure R.2, billions of LTOI SAW filters have been widely used in smartphones like Apple, Samsung and Sony [5-7]. Therefore, the LTOI wafer is the only successful smart-cut-based material platform accepted by the consumer market after the SOI. The continuous maturation and cost reduction of LTOI wafers will be guaranteed by their extensive application in high-performance filters for 5G communications. The market price of the 150mm LTOI wafer has been rapidly reduced to **only 300 USD** in the last 3 years, which is less than 20% of the **LNOI of the same size (> 1500 USD)**.

In the introduction section of our manuscript, **we added an overview of the application context and the industrial landscape surrounding both LNOI and LTOI**. We believe that this contextual

information substantiates our statement that LTOI enjoys a distinct cost advantage when compared to LNOI.

Figure R 2: The cross-sectional image of the IHP-SAW and Ultra-SAW filter devices based on LTOI wafers, which is widely used in smartphone like Samsung, Apple, and Sony [5-7].

- [1] Jia, Y., Wang, L., & Chen, F. (2021) Ion-cut lithium niobate on insulator technology: recent advances and perspectives. Applied Physics Reviews, 8(1).
- [2] <https://www.soitec.com/en/capital-markets-day-2021>.
- [3] Takai, Tsutomu, et al. "IHP SAW technology and its application to microacoustic components." 2017 IEEE International Ultrasonics Symposium (IUS). IEEE, 2017.
- [4] Qualcomm Announces Breakthrough Qualcomm ultraSAW RF Filter Technology for 5G/4G Mobile Devices.
- [5] Soitec announces the addition of piezoelectric-on-insulator (POI) substrate capacity to meet increasing demand for 4G/5G RF filters - Soitec.
- [6] Murata IHP SAW Filter - System Plus Consulting (i-micronews.com).
- [7] Qualcomm UltraSAW technology with Soitec POI substrate finally on market. <https://www.reverse-costing.com/teardown-notes/qualcomm-ultrasaw-technology-soitec-poi-substrate->

Action taken: We added more details in the description of the fabrication of LTOI wafers, and provided a direct comparison of the cost stemming from their differences in preparation recipes.

LNOI has been commercialized and 8-inch lithium niobate wafers have also been implemented. And the LTOI in the text appears to be 4 inches.

Figure R3. (a) and (b) 8-inch LTOI bulk wafers from Sumitomo Electric. (c) 6-inch LTOI thin-film wafers prepared by smart-cut process at SIMIT.

We agree that 8-inch lithium niobate bulk wafers have been demonstrated, but it's important to note that this achievement has also been realized in lithium tantalate. Figure R3(a) and (b) show the 8-inch LiTaO_3 bulk wafers from Sumitomo Electric, demonstrating the availability of 8-inch lithium tantalate (however not bonded to an insulator yet).

The established smart-cut process for the manufacturing of LTOI wafers in this paper is compatible with 6- or 8-inch wafer sizes. However, due to the limitation of wafer grinding and polishing equipment at SMIT, LTOI wafer preparation, the maximum wafer size for our demonstration is currently 6 inches, as shown in Figure R3(c).

In our manuscript, the photonic integrated circuits we presented were fabricated using a 4-inch LTOI wafer. This choice was primarily dictated by constraints within the EPFL university nanofabrication facilities, notably the compatibility of equipment for waveguide dry etching, which currently supports only up to 4-inch wafers. Nonetheless, it is imperative to emphasize that both the smart-cut process and the DUV-lithography-based nanofabrication process are principally adaptable to 6- or 8-inch wafer sizes in a semiconductor manufacturing line. Thus, our work represents a practical and scalable approach for the volume manufacturing of LTOI photonic integrated circuits. Given the superior volume cost scaling and the easier manufacturing process for LTOI wafers, we believe that LTOI can replace LNOI in volume applications.

The advantages of the preparation process are also not significant, comparing to LNOI fabrication as ref 7 in the manuscript.

We respectfully disagree with the reviewer's comment. In comparison to reference 7, we believe that the preparation process shown in this article is of great significance. Reference 7 demonstrated the fabrication of high-density LNOI photonic integrated circuits using DLC as an etching hard mask on the pre-existing and mature LNOI wafer platform. **In contrast, this work establishes a comprehensive process flow that includes efficient LTOI wafer production using the smart-cut method, deep waveguide etching facilitated by a novel cleaning process for LT redeposition from ion milling, and a DUV-compatible thick metal electrodes fabrication process.** These fabrication processes collectively represent significant progress and innovation in the ferroelectric photonics platforms: To clearly elucidate the technological advancements achieved in this work, we want to address each point individually:

- **The Smart-cut process:** Our work introduces a smart-cut process for manufacturing high-quality optical and acoustic grade LTOI wafers with thick bottom oxide for tight optical confinement and optical-RF phase matching that is compatible with 6- or 8-inch wafer sizes. This process utilizes hydrogen ion implantation with low-energy levels (100 keV) to yield high-quality LTOI wafers. As mentioned above, this progress has substantial implications for reducing production costs and enhancing the production efficiency of LTOI wafers compared to LNOI wafers.
- **Deep etching LiTaO₃:** Concerning the specific advantages of deep etching, we would like to emphasize the 'gold standard' in wafer-scale manufacturing of PICs is to etch through the waveguide core layer completely, which does not require timed etching and thus guarantees inherently uniform etch depth across the wafer. Lithium niobate is widely recognized as a hard and relatively inert material, making it challenging to etch. In contrast, LiTaO₃ exhibits not only a higher mass density but also stronger chemical bonds, resulting in increased mechanical and chemical strength. Consequently, etching LiTaO₃ is more challenging compared to LiNbO₃. Therefore, extending the etching method reported in reference 7 from LNOI to LTOI is not a straightforward task. Of particular significance is the fact that due to LiTaO₃'s higher chemical stability, the method of removing by-products with RCA-1 solution, as used in Reference 7, is no longer applicable in the case of LiTaO₃. In our work, we employed a more alkaline solution to effectively clean the by-products. That allows us to deeply etch LiTaO₃ and obtain tightly-confining LiTaO₃ strip waveguides with ultra-low loss.
- **Thick electrode manufacturing based on DUV lithography:** Beyond low-loss optical waveguides, high-quality electrodes are essential for electro-optic devices. To minimize the microwave loss on the electrodes, the fabrication process must be compatible with the deposition of thick metal films, typically exceeding 800 nm in thickness. As detailed in the supplemental material, we employ silicon oxide as a sacrificial layer to facilitate the lift-off process. This innovative approach

represents a novel method for DUV-compatible thick metal preparation, which has not been previously reported.

We emphasize that the progress in the preparation process within this work is the establishment of a comprehensive process flow, which spans from efficient wafer production, through deep waveguide etching, to metallization procedures. The entire nanofabrication process is moreover based on DUV stepper lithography thus rendering it scalable for high-volume manufacturing.

2. Comparing the MZI performance with the LNOI modulators, the modulation efficiency is not prominent. And there seem to be no bandwidth parameters of the LTOI MZI. The low propagation loss and high electrooptical efficiency of LTOI claimed in the manuscript are not clearly reflected.

We thank the Reviewer for pointing out this concern. Regarding the criticisms on the propagation loss, we respectfully disagree that low propagation loss is not reflected in our manuscript. See the performance comparison (cf. Table R 1).

To address the issue of broadband electro-optical modulation, we designed a new MZM device for the demonstration of high-speed and high-efficiency modulator. The designed MZM is composed of two 50:50 adiabatic Y-splitters at either end and a push-pull optical waveguide phase shifter pair with a length of 2.5 mm. To achieve a higher modulation efficiency, the gap between the LT waveguide sidewalls and the gold electrode is designed as 2.5 μm . Additionally, the etching depth of the waveguide is 220 nm, leaving a 400 nm slab, ensuring phase matching of the travelling microwave and optical fields. The measured $V_{\pi}L$ is 1.9 V cm at C-band (cf. Figure. 3(d)-(g)), comparable to or even slightly better than state-of-the-art LNOI modulators (cf. Table R 1) and 1.6 Vcm in the O-band. Fundamentally, the electro-optical modulation efficiency of LT and LN should be similar due to their close electro-optical coefficients ($r_{33} = 30.9$ pm/V in LN and $r_{33} = 30.5$ pm/V in LT).

In our revised manuscript, we added the modulation bandwidth measurement, and the measured 3 dB electro-optical bandwidth is 40 GHz. Although this value is not as high as the state-of-the-art LNOI modulators (typically > 67 GHz), it represents the first high-speed modulator reported in the LTOI platform. We believe that by optimizing the transmission line's design, we can achieve a level similar to that reported in LNOI devices. The reason is that the bandwidth limitation of current Pockels-based modulators doesn't stem from the EO material itself but rather from other factors. These include microwave loss in the transmission line, impedance mismatches with external electrical circuitry leading to reflections and increased voltage, as well as velocity mismatches between the traveling electrical and optical signals.

We emphasize that the novelty of our work lies in confirming that Lithium Tantalate, as a platform has entered a large-volume production stage driven by its applications in 5G filters, which not only shares electro-optical performance similar to LN but also exhibits significantly superior characteristics. These include much lower birefringence (cf. Figure 2) and the ability to generate solitons on its x-cut crystal (cf. Figure 3).

Action taken: We added new measurements of an electro-optic Mach-Zehnder modulator (MZM) with a 3 dB electro-optical bandwidth of 40 GHz and a half-wave electro-optic modulation efficiency of $V_{\pi} \cdot L = 1.9$ V·cm. We reorganised Fig. 3 according to the new measurement.

Fig. 3: Electro-optical tuning and switching in LTOI. (a) Colorized scanning electron micrograph (SEM) of LTOI (blue) racetrack optical microresonator with gold electrodes (yellow). (b) Colorized SEM of pulley resonator and bus waveguide coupling section. (c) Measured resonance shift as a result of tuning voltage. The linear fit indicates a voltage tuning response of 255 MHz/V. Inset left: Schematic of measurement setup for microresonator tuning measurement with phase modulation sideband calibration. Inset right: Electro-optical tuning of LTOI microresonator. Each color step corresponds to an increase in DC tuning voltage of 5 V. (d) Optical micrograph of 2.5 mm long Mach-Zehnder modulator (MZM). (e) Colorized SEM of MZM waveguides and electrodes. (f) Electro-optical bandwidth (S_{21} parameter) of MZM of device length of 2.5 mm at the wavelength of 1550 nm. (g) Normalized optical transmission as a function of applied voltage on traveling wave electrodes at the wavelengths of 1310 nm and 1550 nm, showing a voltage-length product $V_{\pi}L = 1.6$ V cm at O-band and $V_{\pi}L = 1.9$ V cm at C-band.

3. For the LTOI Soliton microcomb, what are the advantages of LTOI compared to other materials (SiN, LNOI ...) to generate soliton combs, especially for LNOI?

We thank the Reviewer's comments. Regarding the process of soliton generation, Si_3N_4 represents the most intensively studied and arguably the most mature platform, benefiting from a low optical loss, a wide transparency window and a CMOS-compatible fabrication process. It is well known that the successful generation of solitons relies on a balance of nonlinear and dispersive effects. In particular, Wang et al. have shown that the Raman nonlinearity places a hard bound on the detuning frequency range of the dissipative Kerr soliton [1]. In comparison between LT and LN, we experimentally find that LT is more conducive to soliton generation in X-Cut substrates. This is supported by our manuscript (cf. Figure 4 and Extended Data Figure 4), demonstrating LT's superior suppression of the Raman effect compared to LN by adjusting the polarization. We will discuss this in detail in response to the reviewer's next comment.

If the reviewer is referring to whether LT has an advantage over SiN in the field of soliton applications, then we believe this advantage is clear. LT exhibits a strong Pockels effect, enabling it to be used for low-loss, ultra-fast frequency modulation or driving—a functionality unachievable with Si_3N_4 due to its lack of fast electro-optical property. Furthermore, the electrically assisted domain-poling technique further enhances LiTaO_3 's potential for controlling nonlinear coupling between fundamental and harmonic modes, making it an ideal platform for on-chip soliton microcomb self-referencing and high-speed actuation. Similarly, LNOI, another ferroelectric material akin to LTOI, shares similar properties. However, LTOI offers the additional advantage of lower birefringence compared to LNOI, making it

superior for generating broadband, dispersion-free soliton via the Kerr effect or electro-optic effect. In addition, our work demonstrated that the Pockels and third-order Kerr nonlinearity can be simultaneously harnessed in X-cut LTOI, which implies it is possible to not only combine high-speed actuation with microcombs but also to combine 2nd and 3rd order nonlinear optical effects in the material.

[1] Wang, Y., Anderson, M., Coen, S., Murdoch, S. G., & Erkintalo, M. (2018). Stimulated Raman scattering imposes fundamental limits to the duration and bandwidth of temporal cavity solitons. *Physical Review Letters*, 120(5), 053902.

The authors may need to study why the soliton microcomb can not be generated in X-cut LNOI, and what is the difference for the soliton microcomb generation between X-cut LTOI and LNOI, and have an overall comparison on those in Z- or X- cut crystal thin films?

We thank the Reviewer for pointing out this concern. We agree with the Reviewer that we need to investigate the disparities in soliton generation experiments conducted on LTOI and LNOI. This analysis will help us understand why we were able to achieve soliton generation on x-cut LTOI, or conversely, why other studies in the literature did not achieve similar results in x-cut LNOI. , The Raman effect is most pronounced when the light is polarized along the crystal's polar Z-axis. In the case of x-cut LNOI, the TE mode encounters the strongest Raman scattering, leading to competition and disruption of soliton formation (see [1] above). This is the main reason why achieving solitons in the x-cut LNOI has proven to be elusive despite extensive research.

Since the Raman phenomenon represents a primary obstacle to soliton generation in both LN and LT, we investigated the polarization-dependent Raman effect in both x-cut LNOI and LTOI, we added the data to Extended Data Figure 4. We observed a reduction in Raman intensity when the polarization of incident light transitions from being parallel to the z-axis to being parallel to the y-axis. Remarkably, this reduction is much more pronounced in LTOI compared to LNOI. **Therefore, in x-cut LTOI, the Raman effect can be suppressed more effectively compared to LNOI by adjusting the polarization of the pump light.** In our work, we use racetrack microresonators with the straight waveguide section oriented along the y-axis. This configuration ensures that the TE mode predominantly aligns with the non-polar y-axis, as depicted in the schematic chip diagram in the manuscript Figure 4.

Regarding the comparison in Z- and X-cut crystal thin films, we currently do not possess samples of z-cut LTOI for direct comparison. However, extensive studies on x-cut LNOI and z-cut LNOI provide valuable reference points. Many efforts have been made to obtain solitons in x-cut LNOI, but they were compounded by the presence of a strong Raman gain that impedes Kerr soliton generation [1,2]. To mitigate the interference of Raman, many works focus on Z-cut LNOI [3-6]. For Z-cut LNOI, the electric field polarization of on-chip TE mode is along the non-polar axis (x or y-axis) of the material. In addition, other strategies also have been employed when attempting to achieve soliton on Z-cut LNOI [3,4].

Finally, we emphasize that the progress of the 30.1 GHz and 80 GHz soliton microcomb generation in x-cut LTOI is innovative and will further advance the application of ferroelectric materials in the microcomb community.

[1] Wang, Cheng, et al. "Monolithic lithium niobate photonic circuits for Kerr frequency comb generation and modulation." *Nature Communications* 10.1 (2019): 978.

[2] Yu, Mengjie, et al. "Raman lasing and soliton mode-locking in lithium niobate microresonators." *Light: Science & Applications* 9.1 (2020): 9.

[3] He, Yang, et al. "Self-starting bi-chromatic LiNbO₃ soliton microcomb." *Optica* 6.9 (2019): 1138-1144.

[4] Gong, Zheng, et al. "Soliton microcomb generation at 2 μm in z-cut lithium niobate microring resonators." *Optics letters* 44.12 (2019): 3182-3185.

[5] He, Yang, et al. "High-speed tunable microwave-rate soliton microcomb." *Nature Communications* 14.1 (2023): 3467.

[6] Gong, Zheng, et al. "Monolithic Kerr and electro-optic hybrid microcombs." *Optica* 9.9 (2022): 1060-1065.

Action taken: We added an introduction on the Raman-induced challenges to explain the reason why the soliton generation in x-cut LNOI remained elusive.

"This Raman interference presents a common challenge when attempting to generate solitons in ferroelectric crystal platforms. For instance, despite extensive research efforts, achieving solitons in the x-cut configuration of LNOI has remained elusive."

We added a statistical analysis/experiments to demonstrate the effectiveness of Raman suppression by orientation.

"In each of 10 devices with the orientation 90° , solitons were consistently generated. Conversely, none of the attempts to generate solitons were successful in the 10 devices oriented at 0° . This demonstrates that altering the orientation to mitigate the Raman effect can be an effective method for generating solitons."

We also added the comparison data about the polarization-dependent Raman intensity in LNOI and LTOI in Supplementary Materials (Extended Data Figure 4).

"It is well known that the Raman effect is polarization-dependent, typically exhibiting maximum strength when the pump light is polarized along the crystal's polar axis. We investigate such polarization-dependent Raman effect in both x-cut LNOI and LTOI (cf. Extended Data Figure 4). A reduction in Raman intensity is achieved when the polarization of incident light transitions from being parallel to the z-axis to being parallel to the y-axis. Remarkably, this reduction is more pronounced in LTOI compared to LNOI. Consequently, in x-cut LTOI, the Raman effect may be more effectively suppressed by adjusting the polarization of pump light than in LNOI."

Extended Data Figure 4: Raman comparison between LNOI and LTOI. Raman spectra of (a) x-cut LNOI and (b) LTOI for the excitation laser with polarization angles 0° , 30° , 60° , 90° with respect to the z-axis. (c) Variation of the Raman intensity ratio of LNOI and LTOI for the excitation laser with different polarization angles. The Raman intensity is obtained by integrating all peaks that correspond to LiNbO_3 or LiTaO_3 and is normalized. Insert illustrates the angles between the orientation and the polarization of the excitation laser.

We also added the detailed orientation information for the racetrack microresonators used for the soliton generation experiment.

“To reduce the Raman interference, we use racetrack microresonators with the straight waveguide section oriented along the y-axis. This configuration ensures that the TE mode predominantly aligns with the non-polar y-axis, as depicted in the schematic chip diagram in Figure 4(c).”

Fig. 4: Dissipative Kerr soliton (DKS) generation in LTOI microresonators. (a) Optical spectrum of a single soliton microcomb featuring a sech²-spectral profile with a 3 dB bandwidth of 4.9 THz corresponding to an FWHM pulse duration of 63 fs at a pulse repetition rate f_{rep} of 81 GHz. The inset shows the generated light during the rapid laser scan measured by filtering out the pump light. (b) Optical spectrum of a single soliton with repetition rate 30.1 GHz. Insert shows the relative phase position inside the microresonator. (c) Optical setup for soliton generation in x-cut LTOI microresonators. Orientation and TE polarization are also indicated in the schematic diagram of the fabricated LTOI chip. Rapid laser scans are generated using a single-sideband modulator (SSB) and an erbium-doped fiber amplifier. The soliton microcombs are analyzed using an optical spectrum analyzer (OSA) and the nonlinearly generated light and microwave beat notes are recorded with a fast photodiode (PD) and analyzed with an oscilloscope (OSC) and electrical spectrum analyzer (ESA), respectively. (d) Variation of the Raman intensity LTOI with different rotation angles. Insert illustrates the angles between the orientation (z -axis) and the polarization of the excitation laser. (e) Optical spectrum of three-soliton state with repetition rate 30.1 GHz. Insert shows three solitons inside the microresonator. (f) Single-side band phase noise power spectral density of 30.1 GHz microwave beat note generated from the multi-soliton state of the panel (e). Inset: Spectrum of microwave beat note with resolution bandwidth 30 Hz.

Referee #2 (Remarks to the Author):

In this manuscript, comprehensive results on the fabrication of various integrated-optical photonic devices (micro-resonators, electro-optical modulators, soliton micro-combs) in the new thin-film material lithium tantalate (LTOI) are presented for the first time in an overview. Compared to the thin-film material LNOI based on lithium niobate, which has been intensively investigated for several years and is basically very similar in terms of optical properties, LTOI exhibits small but important differences. These include higher transparency in the near UV range, higher optical damage thresholds and significantly lower birefringence, which is important for individual optical functions. Previous work dealing mainly with the use of LNOI for photonic devices is appropriately referenced, and the references given clearly support the understanding of the subject. Abstract, the conclusions and summary of the paper are clear and sufficiently informative.

The demonstrated results on photonic integration using the new LTOI platform developed by the authors are very novel, of very high scientific quality and very promising for the development of photonic integrated circuits (PICs); the here presented first results demonstrate already a lot of challenging high-level experimental work. The work submitted here, with a focus also on demonstrating the potential for mass (i.e. wafer scale) production of PICs, has in principle the potential to be highly interesting and valuable to a wider scientific community - beyond the field of photonics - and to lay the foundation for further research efforts on this new optical material platform. The scientific impact of this work is expected to be high. Overall, I would recommend the manuscript for publication in Nature, provided the authors consider the further improvements suggested below.

We thank Reviewer #2 for their careful study of our manuscript, constructive suggestions, and their positive assessment of the novelty and impact of our manuscript. We are happy to provide a detailed response to the raised questions below.

“Some points that should be revised/improved are:

- both abstract and top of page 2: The sentence “We show that LiTaO₃ [...] can be etched with high precision [...] using DUV lithography [...] and alkaline wet etching.” is misleading because in this work LiTaO₃ is dry etched using IBE. Please clarify this part.”

We thank Reviewer #2 for the careful reading. In our nanofabrication, the etching process for the LiTaO₃ PIC involves a combination of dry etching and wet etching, with DLC serving as a hard mask that is precisely defined using DUV lithography.

Action taken: we rephrased the sentence to enhance clarity.

“We show that LiTaO₃ possesses equally attractive optical properties and can be etched with high precision and negligible residues using a combination of dry and alkaline wet etching, established by the use of diamond like carbon (DLC) as a hard mask through DUV lithography.”

in the abstract it is claimed that “LiTaO₃ [...] exhibit higher photorefractive damage threshold.” However, this claim is not supported by any reference. Please either skip this statement or justify it by proving adequate literature sources.

We thank the Reviewer for pointing out our omission of supporting references for this important statement. Indeed, PPLT is often used over PPLN for pulsed frequency conversion or for nonlinear optics in the visible, because of the larger bandgap and higher damage threshold [1].

[1] Eknayan, O., et al. "Comparison of photorefractive damage effects in LiNbO₃, LiTaO₃, and Ba_{1-x}Sr_xTi_yNb_{2-y}O₆ optical waveguides at 488 nm wavelength." Applied physics letters 71.21 (1997): 3051-3053.

Action taken: we added citations to support our statement on photorefractive damage threshold. In addition, we provided adequate references for the statement of loss tangent in the same sentence.

- on top of page 2 it reads “The stronger chemical bonds induce a higher electron density in LiTaO₃”. The meaning of this sentence is not clear for me and seems to be physically incorrect. The number of valence (binding) electrons in LiNbO₃ and LiTaO₃ is identical, although bonds in LiTaO₃ of course are stronger. In total Ta ions have more electrons than Nb ones, but as these are strongly bonded to the cores they do not play a significant role for the bonding strengths or polarizability/refractive index (indeed indices for LiNbO₃ are slightly higher), so an effect of “higher electron density” on material properties is not clear here.

We agree with the Reviewer that there is no strong correlation between chemical bonds strength and electron density. What the reference we cited concludes is that LiTaO₃ possesses stronger bonds, resulting in higher fracture strength and hardness compared to LiNbO₃.

Action taken: We removed the statement: “*The stronger chemical bonds induce a higher electron density in LiTaO₃*” and rephrased the sentence as follows:

“LiTaO₃ is a traditional ferroelectric crystal with a nearly identical crystal structure as LiNbO₃ replacing Nb atoms in the crystal structure with heavier Ta atoms (cf. Figure 1(a)), which makes LiTaO₃ exhibit not only higher mass density but also stronger chemical bonds, thus exhibiting increased strength and chemical stability [20].”

The 2nd part of the sentence says “exhibit not only higher density”; here it is unclear which density (electron or mass?) the authors mean.

Here what we intend to express is that LiTaO₃ exhibits a higher mass density.

Action taken: we rephrased the sentence.

“which makes LiTaO₃ exhibit not only higher mass density but also stronger chemical bonds,”

The given band gap values E_g on top of page 2 (although measured in [18]) are not consistent with most other data from literature.

We thank the Reviewer for pointing out this concern. The determination of the band gap of both LiNbO₃ and LiTaO₃ remains a topic of ongoing debate. There is a large difference in the band gap values of LiNbO₃ and LiTaO₃ reported in the literature. In literature, the experimental band gap values for LiNbO₃ are in the range of 3.3-4.7 eV [1-2], and theoretical estimates can vary from 3.5-6.5 eV [3]. As for LiTaO₃, experimental band gap values are 3.84-3.93 eV [4,5], and theoretical values are 3.7-5.6 eV [6]. The discrepancies in the experimental values typically result from differences in the composition of LiNbO₃ or LiTaO₃ and can also be attributed to variations in measurement methods and conditions. Theoretical band gap values seem to be highly influenced by the calculation methods and input parameters. Among all these various values, the most cited band gap values are 3.93 eV for LiTaO₃ [5] and 3.78 eV for LiNbO₃ [1,7].

Despite the substantial difference in the calculations of band gap values, it is worth noting that the conclusion that the band gap of LiTaO₃ is larger than that of LiNbO₃ is supported by the majority of the literature. Two separate studies [4,5] have compared the band gap values of LiNbO₃ and LiTaO₃ using the same calculation model and testing methods, both concluding that the band gap of LiTaO₃ is larger than that of LN.

Action taken: we replaced the bandgap value of LiNbO₃ with the most cited value 3.78 eV. We cited papers to support our claim that the band gap of LiTaO₃ is larger than that of LiNbO₃.

“The optical band gap of LiTaO₃ (3.93 eV) is larger than LiNbO₃ (3.78 eV) [20,21] “

- [1] Dhar, Ajay, and Abhai Mansingh. "Optical properties of reduced lithium niobate single crystals." *Journal of applied physics* 68.11 (1990): 5804-5809.
- [2] Jiangou, Zhu, et al. "Optical absorption properties of doped lithium niobate crystals." *Journal of Physics: Condensed Matter* 4.11 (1992): 2977.
- [3] Thierfelder, Christian, et al. "Do we know the band gap of lithium niobate?." *Physica status solidi c* 7.2 (2010): 362-365.
- [4] Cabuk, Suleyman. "First-principles study of the electronic, linear, and nonlinear optical properties of Li (Nb, Ta) O₃." *International Journal of Modern Physics B* 24.32 (2010): 6277-6290.
- [5] Çabuk, Süleyman, and Amirullah Mamedov. "Urbach rule and optical properties of the LiNbO₃ and LiTaO₃." *Journal of Optics A: Pure and Applied Optics* 1.3 (1999): 424.
- [6] Huband S, Keeble D S, Zhang N, et al. Relationship between the structure and optical properties of lithium tantalate at the zero-birefringence point[J]. *Journal of Applied Physics*, 2017, 121(2).
- [7] Zanatta, Antonio Ricardo. "The optical bandgap of lithium niobate (LiNbO₃) and its dependence with temperature." *Results in Physics* 39 (2022): 105736.

The composition of the used optical-grade LiTaO₃ is missing – I assume it is congruently melting one. However, in the supplementary section on page 1 the sentence "the optical grade LiTaO₃, which has fewer crystal defects and is closer to the stoichiometric LiTaO₃ crystal" is irritating in this context. Please clarify

We thank Reviewer #2 for this remark. Both optical grade and acoustic grade LiTaO₃ are indeed congruently melting crystals, their composition the same to our knowledge. The difference between them is the defect density. To mitigate the pyroelectric effect, the acoustic grade LiTaO₃ crystal undergoes a reduction reaction enhancing the electrical conductivity. This reduction involves a transformation in the valence state of a portion of the Ta ions, changing from +5 to +4, resulting in a higher concentration of oxygen vacancies within the acoustic LiTaO₃ crystal. The "optical" grade LiTaO₃, which has fewer crystal defects and is closer to the stoichiometric LiTaO₃ crystal" is misunderstanding.

Action taken: we added a sentence to describe the composition of the used optical-grade LiTaO₃. We rephrased the sentence to avoid misunderstanding.

"Note that both optical grade and acoustic grade LiTaO₃ wafers we used are congruent compositions."

"Based on the aforementioned reason, the optical grade LiTaO₃, which has fewer crystal defects, is used in the main manuscript experiments."

page 2: Curie temperature for LiTaO₃: I suggest to write "(610°C – 700°C, depending on the Li/Ta ratio)"

Thank you for pointing out this interesting aspect that the Curie temperature depends on the Li/Ta ratio.

Action taken: we rephrased the sentence according to the Referee's suggestion.

Reference [3] by C. Wang et al. appears again as number [11]

Thank you for pointing out this inconsistency and for the **very careful reading** to improve our manuscript.

Action taken: we removed duplicates in the citation.

the authors highlight several times that they "propose a new solution to remove LiTaO₃ redeposition", but any details on how this is performed are missing on page 3 (it now reads "with an additional wet etch step"). Please add a detailed description of the method that allows to use this method also by others.

Action taken: we added a detailed description of the LiTaO₃ redeposition removal process

The different content of Fig. 1 seems to be not well organized. In my opinion the (tiny) crystal structure can be skipped. The rest of this figure should be organized in a more logical order, i.e. starting with smart cut (f), showing wafer pictures (g,h) and properties (I,j), followed by waveguide fabrication (e) and showing examples of fabricated structures (c,d,b).

We thank the reviewer for the very thoughtful suggestions on how to improve the layout of our figure. We agree with the Reviewer on the suggestion to organise Fig.1 in a more logical order.

Action taken: We organized Fig.1 according to the Reviewer's suggestion, starting with the smart-cut process, showing the wafer picture and properties, followed by fabrication workflow and showing examples of fabricated LTOI photonic building blocks.

Figure 1: Lithium-tantalate-on-insulator (LTOI) substrates and optical waveguides. (a) Schematic of LTOI wafer bonding workflow with H-ion implantation, bonding, splitting, and CMP. (b) Photograph of bonded wafer demonstrating uniform and defect-free bonding. (c) Thickness map of LiTaO₃ thin film on the wafer. (d) Atomic force micrograph of the LiTaO₃ thin film surface. (e) High-resolution STEM image of the LiTaO₃-SiO₂ bonding interface. (f) Schematic of fabrication workflow for LTOI optical waveguides including diamond-like carbon (DLC) hard mask deposition via plasma enhanced chemical vapor deposition (PECVD) from methane precursor, DLC dry etching via oxygen plasma, LiTaO₃ etching via argon ion beam etching (IBE), followed by redeposition and mask removal. LiTaO₃ is illustrated in blue, SiO₂ in purple, DLC in black, and Si in grey. (g) Colorized scanning electron micrograph (SEM) of LTOI microring resonator. (h) Colorized SEM of etched LTOI microring and bus waveguide coupling section. (i) Colorized SEM of etched LTOI (blue) waveguide and sidewall. (j) Colorized SEM cross-section of etched LTOI waveguide on top of SiO₂ bottom cladding (purple).

top of text on page 4: should be Figure 2c (not 3c)

Thank you for the very careful reading and suggestion.

Action taken: we corrected the text according to the Referee's suggestion.

In Fig. 2 part (e), showing a qualitative index ellipsoid (here for negative birefringence, i.e. for LiNbO₃), could be skipped in my opinion as it provides not much information for the reader. The same for part (f), without further details there seems to be no useful information for the reader. Furthermore, this figure part is never referenced in the text.

We thank the Reviewer for pointing out this concern. We respectfully disagree with the suggestion to skip Fig.2 (e) and Fig.2 (f). We believe that these figures enhance the clarity and understanding of birefringence differences between LiTaO₃ and LiNbO₃. In Fig.2 (e), we present a qualitative refractive index ellipsoid for LiTaO₃ and LiNbO₃. This diagram illustrates the change in the refractive index of LiTaO₃ and LiNbO₃ in different directions in the two-dimensional plane, aiding broad readers in understanding the differences between these materials. In Fig.2 (f), we depict how the optical mode undergoes a refractive index change within a waveguide bend. This graphical depiction not only clarifies our simulation results but also defines the relevant angles θ used in Fig.2 (g). These figures may appear intuitive and even simple, but play a crucial role in helping a broad and diverse reader grasp the distinctions in birefringence between LiNbO₃ and LiTaO₃, which is of great importance for applications in integrated photonics. For instance, the low birefringence of LiTaO₃ enables a single LiTaO₃ microresonator to operate across all telecommunication bands. Furthermore, the low birefringence allows for ultra-broadband dispersion engineering of LTOI waveguides and also for electro-optical frequency comb generation, where the bandwidth is limited by dispersion and birefringence and octave-spanning bandwidth has not been achieved to date.

We added additional discussion of the panels in the text and additional discussion of the birefringence.

Action taken: We added a reference to panels e, and f in the text. We also added a deeper discussion of the optical birefringence to the conclusion section.

“Moreover, the low birefringence allows for ultra-broadband dispersion engineering of LTOI waveguides and also for electro-optical frequency comb generation, where the bandwidth is limited by dispersion [15] and birefringence [47] and octave-spanning bandwidth was not achieved to date.”

[47] Hu, Y. et al. High-efficiency and broadband on-chip electro-optic frequency comb generators. Nature Photonics 16, 679–685 (2022)

2nd paragraph of page 4: better write “magnitude of optical birefringence” (because +0.004 is larger than -0.07)

Thank you for the very careful reading and suggestion.

Action taken: we rephrased the sentence according to the Referee's suggestion

Figure 3(f) seems to be not referenced in the text

Thank you for the very careful reading and suggestion.

Action taken: We referenced Figure 3(f) according to the Referee's suggestion

Figure 4(e) seems to be not referenced in the text

We thank the Reviewer for pointing this out.

Action taken: We referenced Figure 4(e) in the revised text.

the first section of the supplementary information is of low quality overall and the necessity to provide all the data in Table 1 is not obvious. As far as optical-grade materials are concerned, the production volume of LiNbO₃ might be far higher than that of LiTaO₃, also costs of such LiNbO₃ wafers are much lower.

We thank Reviewer #2 for their careful study of the material properties table. We updated the table based on the Reviewer’s comments.

While substantial research and application of photonic devices have been conducted using LiNbO₃ (also optical-grade) in the past, the comparison of production volume and cost of bulk materials for LiNbO₃ and LiTaO₃ mentioned by the Reviewer is difficult to make because there is not much difference in the growth processes (Czochralski method for congruently melting crystals) of LiNbO₃ and LiTaO₃. However, when considering thin-film materials like LNOI or LTOI (both acoustic-grade and optical-grade), it becomes evident that LTOI holds a clear advantage in terms of production volume and cost.

In our manuscript, we described the ion-implantation step of the smart-cut process for LTOI as follows “We implanted the hydrogen ion into an X-Cut bulk LiTaO₃ wafer with an energy of 100 keV and a dose of 3.2×10^{16} /cm²”. In contrast, the LNOI smart-cut process, as detailed by Jia, Y. *et al* [1] and widely applied in the production line, requires significantly higher energy, typically ranging from 200-400 keV, and involves helium ions. These differences in ion type and energy have significant implications for production costs and efficiency. Hydrogen ion implantation aligns more closely with the processes used in today's high-volume commercial production of SOI wafers. Regarding implantation energy, commercial ion implanters usually have a maximum working energy of 200 keV, necessitating upgrades, such as additional accelerate tubes, required to reach the energy typically greater than 225 keV required for LNOI fabrication. This implies that existing machinery can be used for LiTaO₃ film fabrication, while customer upgrades are needed for LiNbO₃ film production. As for the production volume, LTOI has entered the consumer electronics market ahead of LNOI driven by its widespread in 5G filters due to its superior temperature stability and low-loss performance. Even though these applications are primarily built upon acoustic-grade LTOI, the cost advantages resulting from the fabrication method and the scale advantage in industrial applications can be expected to extend to optical-grade LTOI as well.

Action taken: We revised the sentence “LiTaO₃ not only enjoys the advantage of higher production volume...”, instead of claiming the production advantages of LiTaO₃, we claim the mass production advantages of LTOI. In addition, we deleted some data such as melting point and thermal properties that are not strongly associated with the topic of this work, and corrected some data based on the Reviewer’s comments.

“LTOI not only enjoys the advantage of higher production volume but also demonstrates comparable or even superior performance when compared to LNOI, owing to the inherent properties of LiTaO₃.”

Table 1: Material properties comparison between LiTaO₃ and LiNbO₃

Properties, congruent	LiTaO ₃	LiNbO ₃
Lattice constant, Å	a = 5.154, c = 13.783	a = 5.148, c = 13.868
Density, g/cm ³	7.45	4.64
Curie point, °C	610	1157
Band gap ^{17,18} , eV	3.93	3.78
Dielectric Constant ^{46,47} @ 100 kHz	$\epsilon_{11,22} = 54, \epsilon_{33} = 43$	$\epsilon_{11,22} = 38, \epsilon_{33} = 28$
Refractive index @ 1550 nm	$n_o = 2.119, n_e = 2.123$	$n_o = 2.21, n_e = 2.14$
Birefringence	$\Delta n = 0.004$	$\Delta n = -0.07$
Electro-optic coefficient ¹ @ 1550 nm, pm/V	$r_{33} = 30.5, r_{51} = 20$	$r_{33} = 30.9, r_{51} = 32.6$
Second nonlinear coefficient @ 1064 nm, pm/V	$d_{33} = -21$	$d_{33} = -27$
Optical damage threshold @ 1064 nm, MW/cm ²	240	120
Coercive electrical field @ kV/mm	21	21

LiTaO₃ not only has some superior properties to LiNbO₃ but also has the disadvantage that its nonlinear coefficient is smaller than that for LiNbO₃, resulting in much lower nonlinear conversion efficiencies. Anyway, the given d₃₃ value for lithium tantalate seems to be wrong; in literature it is typically described as negative with a size of -21 pm/V.

Action taken: we replaced the d₃₃ value of LiTaO₃ with the typically described value of -21 pm/V.

In my opinion it makes no sense to classify optical damage thresholds as "high" or "low" without giving numbers - as far as I know, this has not even been studied quantitatively for LNOI (or LiTaO₃), it is also strongly wavelength dependent.

We thank the constructive opinions from Reviewer #2. A study has investigated optical damage thresholds for LiNbO₃ and LiTaO₃ specifically at a wavelength of 1064 nm [1]. This study revealed that the surface damage threshold for LiNbO₃ was measured at 120 MW/cm², whereas for LiTaO₃, it reached 240 MW/cm². This difference is attributed to the fact that Ta⁵⁺ is more resistant to reduction than Nb⁵⁺ when exposed to high-power laser radiation.

Action taken: We quantitatively describe the optical damage thresholds in the table based on the reported values.

[1] Zverev, Georgii M., et al. "Laser-radiation-induced damage to the surface of lithium niobate and tantalate single crystals." Soviet Journal of Quantum Electronics 2.2 (1972): 167.

The term "poled availability" makes little sense in terms of content; what the authors may mean is the ability to achieve (quasi-) phase matching by periodic poling.

Indeed, what we intend to express is both LiNbO₃ and LiTaO₃ can achieve (quasi-) phase matching through periodic poling. Popular ferroelectric materials can be periodically poled including LiNbO₃, LiTaO₃, and potassium titanyl phosphate (KTP). Among these, LiTaO₃ is one of the earliest materials to be subjected to periodic poling for quasi-phase-matching frequency conversion. Instead of using the term "poled availability", we provide quantitative values of the coercive electrical field required for achieving domain inversion through electrical field poling. For both LiTaO₃ and LiNbO₃, the widely accepted values for the coercive electrical field values are similar at 21 kV/mm.

Action taken: We provide quantitative values of the coercive electrical field for LiNbO₃ and LiTaO₃.

[1] Gopalan, Venkatraman, and Terence E. Mitchell. "In situ video observation of 180 domain switching in LiTaO₃ by electro-optic imaging microscopy." Journal of Applied Physics 85.4 (1999): 2304-2311.

[2] Myers, Lawrence Edward, et al. "Quasi-phase-matched optical parametric oscillators in bulk periodically poled LiNbO₃." JOSA B 12.11 (1995): 2102-2116.

Birefringence is defined as $\Delta n = n_e - n_o$, i.e., it is negative for LiNbO₃ (please correct this in table 1).

We thank the Reviewer for pointing this out.

Action taken: We corrected the birefringence value for LiNbO₃.

The lower transparency limit for LiNbO₃ is more like ~350nm instead of the value of 400nm given now. It should be added to Table 1 that all parameters apply to the congruently melting compositions. The band-gap values given in table 1 (from [18], obtained for an absorption coefficient of 300cm⁻¹ for not further specified crystals/compositions) are not consistent with most other literature data and are in disagreement with the given lower transparency ranges/wavelengths. A good overview for LiNbO₃ can be found in: Results in Physics 39, 105736 (2022).

We thank the Reviewer for the comments regarding the band gap and its associated transparent edge wavelength. As we replied above, the determination of the band gap of both LiNbO₃ and LiTaO₃ remains

a topic of ongoing debate, but the most cited band gap value is 3.93 eV for LiTaO₃ and 3.78 eV for LiNbO₃. The literature [Results in Physics 39, 105736 (2022)], as mentioned by Reviewer #2, also supports this perspective. Therefore, we have updated the bandgap value for LiNbO₃ to 3.78 eV, in contrast to the previously mentioned value of 3.63 eV. In this context, it becomes evident that the provided narrower transparency ranges/wavelengths in Table 1 do not disagree with the revised bandgap values. Furthermore, the broad transparent window spanning 0.28-5 μm of LiTaO₃, as claimed in the early literature [Meyn, J-P., and M. M. Fejer. Optics letters 22.16 (1997)], lacks specific details about the composition of LiTaO₃ and the measurement condition. Therefore, to avoid misleading, we deleted the numerical comparison between LiNbO₃ and LiTaO₃ on transparent windows in the table.

Action taken: We replaced the bandgap value of LiNbO₃ with the most cited value 3.78 eV. We deleted the numerical comparison between LiNbO₃ and LiTaO₃ on transparent windows in Table I. We added the statement that all parameters in Table 1 apply to the congruently melting compositions. We cited the literature [Results in Physics 39, 105736 (2022)].

“It should be noted that all the properties listed in the table correspond to the congruent compositions, which are both more readily available and more widely used across various fields compared to their stoichiometric counterparts.”

In the section on linear losses of the supplementary material the fabricated spiral is claimed first to have a length of 39cm, while later in the text it has 90cm length. Please check this.

The fabricated spiral has a length of 39 cm, as evident from the optical microscopy image shown in Figure 3(a). Because our optical frequency domain reflectometer is calibrated to an optical frequency comb, we measure distance not in relation to a fiber reference cavity but our system measures the optical length directly. The group index of the waveguide is 2.25, so the optical distance can be calculated as 87.75 cm. Such optical distance also can be measured from the OFDR signal as depicted in Figure 3(b)-(g). In Figure 3(e) and (f), we enlarge two distinct sharp reflection regions at 3052 mm and 3932mm, showing the successful identification of the front and back facets of the chip. Therefore, the optical distance can be further confirmed as 88 cm. We thank the reviewer for this comment, we have modified the optical distance of the waveguide spiral chip to 88 cm.

Action taken: We checked both values, and we modified the optical distance of the waveguide spiral chip to 88 cm.

Reviewer Reports on the First Revision:

Referees' comments:

Referee #1 (Remarks to the Author):

The authors demonstrated results on photonic integration using the new LTOI platform developed by the authors are promising for the development of photonic integrated circuits (PICs) based on LTOI. It is novel especially for the optical-grade LiTaO₃ wafer fabrication in the manuscript. But for the electro-optical modulation and soliton microcomb generation based on LTOI, the results in the article do not reflect the greater advantages of LTOI compared to other platforms that claimed in the manuscript. So we don't think this paper meet the publication criteria for Nature's or even the journal of "OPTICA", such as "high technical quality" or "high-impact results". The detailed comments are provided as follows.

Firstly, for the LTOI etching, the method used in the manuscript is very similar to the LNOI etching method. Of course, the method is useful and innovative, but it is not the advantage of the LTOI in the manuscript. (Li, Z., Wang, R.N., Lihachev, G. et al. High density lithium niobate photonic integrated circuits. Nat Commun 14, 4856 (2023).)

Secondly, for the LTOI electro-optical modulation, the authors listed the comparison of modulation parameters between lithium tantalate and lithium niobate modulators to prove the superiority of LTOI. However the bandwidth of the modulator are not given and as far as I am concerned, the EO efficiency of the LNOI modulators is better than the LTOI modulator of the manuscript. ($V\pi.L_{0.8V.cm-100GHz}$, Vol. 9, No. 1 / January 2022 / Optica)

Moreover, the key metrics for EO modulators include the half-wave voltage ($V\pi$), EO bandwidth (BW), and insertion loss (IL), where the insertion loss includes linear optical loss or the propagation loss, metal absorption loss and insertion loss of splitters/combiners. Many works would give insertion loss rather than propagation loss. Table R 1 is even not able to complete the comparison of the EO performance in dynamic condition.

Please see the following references to give an objective and correct judgement.

(Chen G, Gao Y, Lin H L, et al. Compact and Efficient Thin-Film Lithium Niobate Modulators[J]. Advanced Photonics Research, 2023, 4(12): 2300229.

Zhu D, Shao L, Yu M, et al. Integrated photonics on thin-film lithium niobate[J]. Advances in Optics and Photonics, 2021, 13(2): 242-352.)

Thirdly, for the soliton microcomb generation the authors did not clarify the advantages of the LTOI compared to SiN or LNOI, and just emphasize the Si₃N₄ material is amorphous and lacks a Pockels coefficient. However the soliton microcomb generation of LTOI in this manuscript did not use the LTOI Pockels effects.

Fourthly, in the submitted manuscript and the response letter, the author has repeatedly emphasized the low birefringence of lithium tantalate and compared it with lithium niobate. Indeed, this property can promote the applications of LTOI in all-band communication. However, its

importance should not be as pronounced as the author suggests in this paper.

1. Firstly, C-band is still the mainstream working band for the integrated photonics because of the relatively-low absorption loss, availability of the inexpensive light sources and high-power fiber amplifiers [1]. As the author has indicated in the main text, for a common LNOI with a film thickness of 600 nm, no mode-mixing will occur in the C-band and even the whole S-band and a part of the E-band, which means in the most conditions, effect from the birefringence in fundamental mode-mixing can be omitted. Moreover, the author has chosen a relatively large etching depth of 500 nm in the simulation. According to the theoretical study in previous work, the threshold frequency of mode-mixing can be further increased if we use the shallow etching [2].

2. Secondly, even if mode mixing is unavoidable in the waveguide, its effect can be weakened by adding a cladding layer [2] and cancelled out by the design of the radius of the bending waveguide [3]. It is worth noting that in this case the strength of mode mixing is insensitive to a larger or smaller radius, so that compactness can be ensured at the same time.

3. At last, the view of birefringence in this work is quite one-sided. From another perspective, huge birefringence induced mode-mixing on X-cut LNOI can be used to develop the polarization-insensitive devices [4] that is highly desired in the optical communication networks. In addition, huge-birefringence itself could also enable the high-efficiency and broadband nonlinear conversion in a poling-free configuration [5,6]. However, these applications are blocked for the LTOI due to the limited birefringence.

4. Please see these references as follows to support the above comments.

References

[1] Boes, Andreas, et al. "Lithium niobate photonics: Unlocking the electromagnetic spectrum." *Science* 379.6627 (2023): eabj4396.

[2] Wang, Jingyi, et al. "Polarization coupling of X-cut thin film lithium niobate based waveguides." *IEEE Photonics Journal* 12.3 (2020): 1-10.

[3] Cheng, Jian, et al. "Ultra-efficient second harmonic generation via mode phase matching in integrated lithium niobate racetrack resonators." *Optics Express* 31.22 (2023): 36736-36744.

[4] Li, Xuepeng, et al. "Monolithic 1×4 Reconfigurable Electro-Optic Tunable Interleaver in Lithium Niobate Thin Film." *IEEE Photonics Technology Letters* 31.20 (2019): 1611-1614.

[5] Lu, Chuanyi, et al. "Highly tunable birefringent phase-matched second-harmonic generation in an angle-cut lithium niobate-on-insulator ridge waveguide." *Optics Letters* 47.5 (2022): 1081-1084.

[6] Lin, Jintian, et al. "Phase-Matched Second-Harmonic Generation in an On-Chip LiNbO₃ Microresonator." *Physical Review Applied* 6.1 (2016): 014002.

What's more, the authors emphasize that, Lithium Niobate wafers are costly and not available in high volumes. Actually, optical-grade LTOI are also costly according to what I know today, as for LiTaO₃ adopted for 5/6G wireless filters, LNOI also can be used as a SAW filters and also commercialized already; and the production of optical-grade LTOI wafers are no mass production, On the contrary, the price of LNOI is gradually decreasing.

Optical-grade LiTaO₃ wafer fabrication in the manuscript and the photonic devices fabricated in the LTOI firstly are novel, but the photonic devices are extensively researched in other materials platform and the experiment result in the manuscript not reflecting the author's claimed advantag

Referee #2 (Remarks to the Author):

The manuscript has been revised very thoroughly and comprehensively, which has improved the quality of the content even further. The points I made in the first review have been fully addressed. The authors also address the criticism expressed by Reviewer #1 thoroughly and, in my opinion, appropriately and convincingly. Overall, I recommend the paper in its present form for acceptance by Nature.

**Resubmission of a revised version of
Nature Manuscript Number 2023-06-11067**
“Lithium tantalate photonic integrated circuits for volume manufacturing”

We are grateful that two reviewers have seen our manuscript. After reading the comments made by the referees thoroughly, we would like to thank them for the detailed review.

We were pleased to read that Reviewer #2 recommended our work for publication and also recognized the thoroughness and convincingness of our response to the criticism from Reviewer #1 in the first round of review.

Despite the positive comments from Reviewer #2, we are concerned by the reply to our comments of Reviewer #1. In the 1st round of review, we have addressed comments raised by Reviewer #1. We have added measurements of the LTOI modulator bandwidth, and included measurements regarding the Raman nonlinearity, comparing spectra at different crystal orientations for LiNbO₃ and LiTaO₃, which explain the feasibility of soliton microcomb generation in an X-Cut wafer crystal. To address the cost concerns raised by Reviewer #1, we provided a detailed comparison between LNOI and LTOI in terms of the ion-implantation process and the current status regarding their mass production. In this round of review, Reviewer #1 repeated similar questions we responded to in the 1st round and raised new points such as the advantages of low birefringence and MZM insertion loss. We strongly disagree with several of the criticisms raised by Reviewer #1. Below, we provide a point-by-point response to the reviewers' comments and questions. The **original review report is printed in black, our responses in blue, and the action taken in red.**

Referee #1 (Remarks to the Author):

The authors demonstrated results on photonic integration using the new LTOI platform developed by the authors are promising for the development of photonic integrated circuits (PICs) based on LTOI. It is novel especially for the optical-grade LiTaO₃ wafer fabrication in the manuscript.

We thank the reviewer for his/her comments on the novelty.

But for the electro-optical modulation and soliton microcomb generation based on LTOI, the results in the article do not reflect the greater advantages of LTOI compared to other platforms that claimed in the manuscript. So we don't think this paper meet the publication criteria for Nature's or even the journal of "OPTICA", such as "high technical quality" or "high-impact results".

We are surprised by the comment concerning the novelty and firmly disagree with the criticism that the advantages of LTOI are not established, in particular for the two quoted applications of 'electro-optical modulation' and 'soliton microcomb' generation.

Our results clearly reflect the advantages of LTOI over LNOI across many aspects. This includes first and foremost the innate material properties, that may benefit many other applications of ferroelectrical PICs and that have been documented in literature.

- **A higher photorefractive damage threshold and a lower photorefractive effect in comparison to LN (see additional measurements at the end of this reply and in the updated manuscript)**
- **Lower microwave loss tangent**
- **Larger bandgap compared to LN for UV operation**
- **Lower birefringence of LT compared to LN**
- **The capability of generating Dissipative Kerr solitons in an X-cut crystal thin film**

For electro-optic modulators, which is a widely anticipated 'volume' application in interconnects for datacenter the price per chip is critical. The EO constant (r_{33}) of both materials (LN and LT) is nearly identical, implying equally efficient modulation. Yet, for this market to adopt electro-optical PICs the availability of cheap substrates is critical. The latter is already the case for LT but not for LN.

The reasons why the cost of the LTOI is and will be significantly lower than LNOI:

First, LTOI intended for filter applications is already in mass production today (750k wafers/year), and the facilities are fully compatible with transitioning to LTOI which is aimed for optical applications, requiring only a simple switch of the bulk raw materials without increasing manufacturing costs. Actually, the raw LT wafer is the same as the wafer used in the mass production of RF filters. The only difference between the optical LT and acoustic LT is the additional reduction process that acoustic (RF) wafers undergo, which is needed for the safe operation of the acoustic filters by mitigating the pyroelectric effect. In fact, the cost of the optical-grade LT wafer without the extra manufacturing step of thermal reduction is even lower than for the acoustic grade LT wafer. There is no difference in bonding performance and yield between the two either.

Secondly, the dominant cost occurs during the film fabrication. The price of the bulk LN wafer is around 100 US dollars which is only 5% of the price of LNOI around 2000 US dollars, therefore, the efficiency of the ion implantation for ion-slicing is the bottleneck and determines the final cost of the LNOI, ion-slicing of LN can be only realized via high-energy Helium ion implantation due to the LN-H bonding issue.

This is a problem since most ion implantation machines are geared towards low energy H-ion implantation for SOI fabrication. Hence, the fact that **H implantation can be used for ion-slicing of**

LT is a significant difference. The current of the H ion beam is generally one order of magnitude higher than the Helium ion beam according to most commercial ion implanters, e.g., GSD 200E. This is related to the fact that the ionization efficiency of H ($I = 13$ eV) being much higher than the noble gas He ($I = 24$ eV) gas. Moreover, for ion-slicing of the standard 500 nm thin film, 180 keV He for LN and 80 keV H for LT are required respectively, since the He ion is heavier than the H ion. Therefore, in contrast to the high cost of LNOI, the cost of a 6-inch LTOI is around 300 US dollars and approaching the SOI wafer cost.

According to the Soitec and NSIT, **LTOI is the second OI wafer successfully applied in mass production** (about **0.2 million in 2023** and 0.7 million wafers predicted in 2024) after the ubiquitous SOI technology and accepted by many major manufacturers of 5G cellular signal filters [1-5].

In conclusion, LT photonics will be the ultimate competitor of Si photonics due to the comparable price of the OI wafer and the inherent performance advantages of LT. In fact, the only reason that LTOI was not adopted for bulk optical modulators has been the failure to manufacture waveguides by Titanium ion-diffusion in the 1980s due to the lower Curie temperature of 600°C.

Concerning the second application, and specifically for **optical frequency comb** generation, the advantages of LTOI include:

- The ability to generate octave-spanning **electro-optical** frequency combs (due to reduced mode crossings for zero-dispersion waveguide geometry)
- The ability to generate and access soliton microcomb (Dissipative Kerr solitons) in the X-Cut crystal orientation (as shown in the manuscript), which was not possible in X-cut LNOI due to the strong Raman nonlinearities.

To provide a clear overview of the advancements achieved in our work, we summarize them as the following:

- We demonstrate a smart-cut process for producing high-quality LTOI using **H⁺ ion implantation with implantation energy significantly lower** than that used for LNOI wafer preparation. This process has substantial implications for reducing production costs and enhancing the efficiency of LTOI wafer manufacturing compared to LNOI wafers.
- We demonstrate for the first-time propagation losses on par with LNOI at 1550 nm and develop for the first-time etching techniques for low-loss LTOI integrated waveguides.
- We demonstrate for the first time LTOI MZM with a high electro-optical modulation efficiency (1.9 V*cm) and a large bandwidth (40 GHz) at 1550 nm.
- Figure 3e in our manuscript reflects that the LTOI platform can process signals across all optical communications bands (**1260-1620 nm**) in a single photonic integrated circuit without mode mixing, which is not achievable with LNOI due to the higher birefringence. This could enable future broadband EO combs.
- We demonstrate the advantage of LTOI in suppressing the Raman effect, enabling us to achieve **soliton microcomb generation for the first time in an x-cut ferroelectric platform**. This enables microcombs with efficient electro-optical modulation (in contrast to Z-cut which requires out-of-plane electrodes).

We disagree with the reviewer on meeting the publication criteria such as “high technical quality” and “high-impact results”:

High technical quality: We develop photonic integrated circuits based on lithium tantalate-on-insulator (LTOI) wafers. We demonstrate low optical loss and tightly confining Lithium Tantalate strip waveguides and demonstrate their application for electro-optic modulators. Similar to LiNbO₃, LiTaO₃

is difficult to etch and we developed a **novel recipe that enables the removal of LiTaO₃ sputter re-deposition and enables low loss waveguides with 5.6 dB/m loss**. In addition, we harness the strong Pockels and Kerr nonlinearities of LiTaO₃ and demonstrate fast electro-optical modulation and soliton microcomb generation in X-cut LTOI. This implies it is possible to combine soliton microcomb generation and high-speed electro-optical modulation in a single photonic platform. This is noteworthy as prior work on LNOI could achieve soliton generation, but only in z-cut and not in x-cut LNOI, which compounds the seamless integration of the soliton microcomb generators and electro-optical modulators, due to the out-of-plane electrical field. We demonstrate experimentally, that LTOI integrated photonics exhibit significantly lower birefringence, allowing devices that operate mode-crossing-free over all currently employed telecommunication bands (from O- to the L-band).

High-impact result: We believe that the future widespread application of LTOI in the optical field will start the pioneering exploration in LTOI, given the physical advantages of the material, and the prospect of much lower wafer costs. In addition, our results are not only of scientific importance but also crucial to the future adoption of electro-optical photonic integrated circuits. Our work paves the way for scalable manufacturing of low-cost and large-volume electro-optical photonic integrated circuits, using a platform that is ramping up production for 5G/6G filters.

[1] <https://www.soitec.com/en/capital-markets-day-2021>.

[2] Takai, Tsutomu, et al. "IHP SAW technology and its application to microacoustic components." 2017 IEEE International Ultrasonics Symposium (IUS). IEEE, 2017.

[3] Qualcomm Announces Breakthrough Qualcomm ultraSAW RF Filter Technology for 5G/4G Mobile Devices.

[4] Soitec announces the addition of piezoelectric-on-insulator (POI) substrate capacity to meet increasing demand for 4G/5G RF filters - Soitec.

[5] Qualcomm UltraSAW technology with Soitec POI substrate finally on market.

<https://www.reverse-costing.com/teardown-notes/qualcomm-ultrasaw-technology-soitec-poi-substrate->

Action taken: We have revised the description of the ion implantation process in LTOI fabrication within the main text to highlight the higher fabrication efficiency achieved through the use of hydrogen ions, which require a lower implantation energy (100 keV) and have a higher beam current, compared to the traditional LNOI fabrication process that uses helium ion implantation with energies exceeding 200 kilo electronvolts.

“In contrast to the well-established LNOI preparation process that utilizes helium ion implantation with an energy higher than 200 keV, the fabrication of LTOI favors hydrogen ions with half the implanted energies 100 keV and beam current ten times higher, as found in most commercial ion implanters, simplifying the wafer production.”

Firstly, for the LTOI etching, the method used in the manuscript is very similar to the LNOI etching method. Of course, the method is useful and innovative, but it is not the advantage of the LTOI in the manuscript. (Li, Z., Wang, R.N., Lihachev, G. et al. High-density lithium niobate photonic integrated circuits. Nat Commun 14, 4856 (2023).)

We respectfully disagree with the reviewer’s comment.

The reviewer did not make a substantive comment on the points we responded to in the first round but insisted on the criticism of the similar etching method we used in the manuscript compared with the etching method used in LNOI. (Li, Z., Wang, R.N., Lihachev, G. et al. High density lithium niobate photonic integrated circuits. Nat Commun 14, 4856 (2023)). We note that while ion beam etching is used as for nearly all ferroelectrics to date, it has never been reported for LTOI thin films. We emphasize again that the typical fabrication recipes used in LNOI do not work for LTOI. We introduce a new recipe for the **removal of LiTaO₃ redeposition, which is a critical step to achieve low sidewall roughness**.

It is well known that attempts to etch LN or LT with Ar sputtering produce by-products which are non-volatile and ultimately attach to the etched sidewalls increasing roughness. LiTaO_3 exhibits not only a higher mass density but also stronger chemical bonds, resulting in increased mechanical and chemical strength of the etch residue. The method of removing by-products with RCA-1 solution, as used in Reference 7, **is no longer applicable in the case of LiTaO_3** . In our work, we hence **developed a novel method** employing a more alkaline solution ($\text{KOH}+\text{H}_2\text{O}_2$) to effectively clean the by-products, which is a **completely new method** and is not similar to the LNOI etching method that has been reported.

Action taken: We have emphasized that the typical removal procedure of LTOI based by-products does not follow the same recipe as for LNOI. We added the sentence in the main text and Methods: *“We find, as detailed in Methods, that the removal of non-volatile by-products requires a different chemical, than for LNOI”*

“After dry etching, it is well-known that in the LNOI case, an additional wet etching step with RCA-1 solution is needed because of the non-volatile by-product accumulating on the waveguide sidewall. However, LT exhibits not only a higher mass density but also stronger chemical bonds (see Extended Data Table), resulting in increased mechanical and chemical strength, the method of removing by-products is no longer applicable in the case of LT. Here, we remove the LT redeposition with a more alkaline solution ($\text{KOH } 40\%:\text{H}_2\text{O}_2 \text{ } 30\% = 3:1$).”

Secondly, for the LTOI electro-optical modulation, the authors listed the comparison of modulation parameters between lithium tantalate and lithium niobate modulators to prove the superiority of LTOI. However the bandwidth of the modulator are not given and as far as I am concerned, the EO efficiency of the LNOI modulators is better than the LTOI modulator of the manuscript. ($V_\pi \cdot L \sim 0.8 \text{V}\cdot\text{cm}$ -100GHz, Vol. 9, No. 1 / January 2022 / Optica)

We would like to clarify that we never claimed that LTOI is superior in modulation compared to LNOI. The statement we have in the manuscript is that LTOI has a comparable electro-optical modulation with LNOI, their similar performance is guaranteed by their electro-optical coefficients ($r_{33} = 30.9 \text{ pm/V}$ in LN and $r_{33} = 30.5 \text{ pm/V}$ in LT).

The reviewer missed the new measurement data that we provided in the first revised manuscript showing the electro-optical modulation spectrum. Our Mach-Zehnder LT modulator exhibits a 3 dB electro-optical bandwidth of 40 GHz.

Regarding the EO efficiency comparison between LNOI and LTOI, in the LNOI platform mentioned by the reviewer, $V_\pi \cdot L$ reaches $0.8 \text{ V}\cdot\text{cm}$ only at 784 nm [1]. Comparing the $V_\pi \cdot L$ at the 748 nm wavelength of the LNOI platform with the $V_\pi \cdot L$ at the 1550 nm wavelength of our LTOI platform is incorrect as $V_\pi \cdot L$ is naturally smaller in the short wavelength range because the same refractive index modulation over the same length will lead to a stronger phase modulation for short wavelength fields. Moreover, the shorter evanescent field penetration into the waveguide cladding facilitates a smaller electrode gap. Typical $V_\pi \cdot L$ values for LN at 1550 nm are in the region of 2 V, commensurable with our results.

[1] Valdez, Forrest, Viphretuo Mere, and Shayan Mookherjea. "100 GHz bandwidth, 1 volt integrated electro-optic Mach–Zehnder modulator at near-IR wavelengths." Optica 10.5 (2023): 578-584.

Action taken: To better clarity, we have given both r_{33} values quantitatively in the main manuscript and Extended Data Table 1. In addition, we have cited the best results in LNOI to compare with previous literature on the $V_\pi L$ in the revised version:

“The $V_\pi L$ of 1.9 cmV is similar to the state-of-the-art results in LNOI with similar electrode structures at 1550 nm, “..., given the fact that LiNbO_3 and LiTaO_3 have almost identical Pockels coefficients between LiNbO_3 and LiTaO_3 (Extended Data Table 1).”

Moreover, the key metrics for EO modulators include the half-wave voltage (V_π), EO bandwidth (BW), and insertion loss (IL), where the insertion loss includes linear optical loss or the propagation loss, metal absorption loss and insertion loss of splitters/combiners. Many works would give insertion loss rather than propagation loss. Table R 1 is even not able to complete the comparison of the EO performance in dynamic condition. Please see the following references to give an objective and correct judgement. Chen G, Gao Y, Lin H L, et al. Compact and Efficient Thin-Film Lithium Niobate Modulators[J]. Advanced Photonics Research, 2023, 4(12): 2300229. Zhu D, Shao L, Yu M, et al. Integrated photonics on thin-film lithium niobate[J]. Advances in Optics and Photonics, 2021, 13(2): 242-352.)

We had provided half-wave voltage (V_π) and EO bandwidth in our 1st revised manuscript (Figure 3f). We emphasize that our work is not about achieving a record performance modulator that exceeds state-of-the-art devices in all parameters. The new point raised by the reviewer is insertion loss. Of course, the total insertion loss for a modulator does include the splitters. Although the insertion loss is crucial in the modulator product in practical applications, it is unreasonable for the reviewer to criticize us based on the insertion loss for the first modulator fabricated in LTOI. Regarding the performance of splitters and fiber couplers, we foresee those little differences will exist between LT and LN at similar levels of maturity. We emphasize that the application case for LN and LT waveguides goes well beyond simple electro-optical modulators and the propagation loss is the key metric that defines the application potential of many aspects. The insertion loss that includes coupling loss and on-chip loss can be optimized but again this is not the main point here and little differences exist between LT and LN.

Regarding the dynamic performance, according to the optical transmission spectrum shown in Fig. 3(g), the measured extinction ratio is 15 dB, we added this value to the manuscript. Other dynamic EO metrics, such as third-order intermodulation distortion (IMD3) and spurious-free dynamic range (SFDR) are important when applying the modulator to real-world applications, however, we believe they are out of the scope of our work.

In fact, recent review literature [1] mentioned by the Reviewer does not include any comparison of insertion loss and dynamic range performance.

[1] Chen G, Gao Y, Lin H L, et al. Compact and Efficient Thin-Film Lithium Niobate Modulators[J]. Advanced Photonics Research, 2023, 4(12): 2300229.

Action taken: we added the measured extinction ratio in the revised manuscript.

“The extinction ratio is measured to be 15 dB and could be further improved by using directional couplers.”

Thirdly, for the soliton microcomb generation the authors did not clarify the advantages of the LTOI compared to SiN or LNOI, and just emphasize the Si₃N₄ material is amorphous and lacks a Pockels coefficient. However, the soliton microcomb generation of LTOI in this manuscript did not use the LTOI Pockels effects.

Our work shows a new capability of generating soliton microcombs (Dissipative Kerr solitons) in LTOI. This represents a significant advancement as in numerous investigations in the LNOI platform, the generation of soliton in the X-cut configuration has been acknowledged as a considerable challenge. Our research demonstrates that solitons can indeed be generated using the X-cut LTOI, which implies it is possible to combine high-speed actuation with microcombs. A recent example of this problem is documented by Song et al. [1], who needed to use two separate photonic chips to generate combs (Z-

Cut) and to efficiently modulate them (X-Cut). Using Z-Cut LN for the modulation instead would have entailed a 10 dB microwave power penalty.

As for the advantages of LTOI, the electro-optical capability of LT is a key advantage. Our previous work on frequency-modulated continuous wave (FMCW) LiDAR based on Si₃N₄-LN heterogeneous platform [2] and piezoelectric control of soliton microcombs [3] are two example applications that can benefit.

Our paper is not about demonstrating every new device that can result from it. We do politely but strongly disagree that such a demonstration should be part of the first paper that reports a *new platform*.

[1] Song Yunxiang, et al. "Hybrid Kerr-electro-optic frequency combs on thin-film lithium niobate", arXiv:2402.11669 (2024)

[2] Snigirev, Viacheslav, et al. "Ultrafast tunable lasers using lithium niobate integrated photonics." Nature 615.7952 (2023): 411-417.

[3] Liu, Junqiu, et al. "Monolithic piezoelectric control of soliton microcombs." Nature 583.7816 (2020): 385-390.

Fourthly, in the submitted manuscript and the response letter, the author has repeatedly emphasized the low birefringence of lithium tantalate and compared it with lithium niobate. Indeed, this property can promote the applications of LTOI in all-band communication. However, its importance should not be as pronounced as the author suggests in this paper.

We thank the reviewer for this comment. We have simply listed this property of LTOI, contrasted it with LNOI, and made the reader aware of the mode-crossings-free phenomenon we observed in our work.

Action taken: we removed the sentence "The latter is particularly important as it allows manufacturing of tightly confining waveguides with sharp bends without mode mixing, that can operate across all telecommunication bands simultaneously (from O to L band)."

1. Firstly, C-band is still the mainstream working band for the integrated photonics because of the relatively-low absorption loss, availability of the inexpensive light sources and high-power fiber amplifiers [Boes, Andreas, et al. "Lithium niobate photonics: Unlocking the electromagnetic spectrum." Science 379.6627 (2023): eabj4396.]. As the author has indicated in the main text, for a common LNOI with a film thickness of 600 nm, no mode-mixing will occur in the C-band and even the whole S-band and a part of the E-band, which means in the most conditions, effect from the birefringence in fundamental mode-mixing can be omitted. Moreover, the author has chosen a relatively large etching depth of 500 nm in the simulation. According to the theoretical study in previous work, the threshold frequency of mode-mixing can be further increased if we use the shallow etching [Wang, Jingyi, et al. "Polarization coupling of X-cut thin film lithium niobate based waveguides." IEEE Photonics Journal 12.3 (2020): 1-10.].

We thank the reviewer for this comment. Birefringence-induced mode-crossing is indeed considered a significant issue in LNOI photonics, with the wavelength varying depending on the waveguide structure. Our paper specifically addresses deep etching situations. However, according to the theoretical study [1] mentioned by the reviewer, in many common configurations with half-etching and SiO₂ cladding, mode-mixing occurs in S-band and even in C-band [2], thus it cannot be ignored. Especially in the situation of broadband nonlinear conversion [3], mode-mixing, regardless of the band it occurs in, can have a significant impact and needs to be avoided. LTOI does not have this problem.

As an example, in the pursuit to achieve broadband electro-optical frequency comb in LNOI, despite overall good performance, the bandwidth was limited by dispersion and birefringence [3, 4], and to date, octave-spanning EO combs have not been achieved precisely for this reason. Recently, our experiment showed that the low birefringence of LTOI can overcome this challenge. In our 1st response letter, we showed the simulated dispersion in a deep-etched LTOI microresonator with the dimension of 700 nm x 2500 nm, where a flat dispersion can be achieved in simulations. In this reply, we added experimental results with good matching between simulations and measurements, as depicted in Figure R1 (our latest result, not shown in this manuscript). Although the demonstration of an octave-spanning EO comb is still underway, our current findings suggest that achieving an octave-spanning EO comb in LTOI is very likely. This is particularly significant because an octave-spanning EO comb has never been demonstrated before and is useful for optical frequency division for the generation of low noise microwaves.

Figure. R1 Comparison between LTOI and LNOI [4] for waveguide dispersion engineering and tailored EO comb phase-matching condition. Blue dots represent the experimental characterization and light blue line simulation results for the LT microresonator from EPFL. The red curve is the dispersion curve for devices used in M. Loncar group for EO combs.

Regarding mitigating the mode-crossings by some special approach in LNOI, we agree that appropriate structural design can circumvent this issue in LNOI, however not without drawbacks such as strong anomalous dispersion or higher waveguide losses. The theoretical study [1] mentioned by the reviewer **was also cited** in our manuscript. However, the birefringence-induced mode-crossing indeed limits the design flexibility and compactness of LNOI PICs in many cases. Trade-offs such as the use of uncladded waveguides or very shallow etching combined with large bending radii may be undesirable for a variety of reasons. Figure R1 is an example of LTOI having more design flexibility to meet both dispersion and birefringence requirements in a manner that is simply inaccessible using LNOI.

[1] Wang, Jingyi, et al. "Polarization coupling of X-cut thin film lithium niobate based waveguides." IEEE Photonics Journal 12.3 (2020): 1-10.

[2] Pan, An, et al. "Fundamental mode hybridization in a thin film lithium niobate ridge waveguide." Optics Express 27.24 (2019): 35659-35669.

[3] Hu, Yaowen, et al. "High-efficiency and broadband on-chip electro-optic frequency comb generators." Nature Photonics 16.10 (2022): 679-685.

[4] Zhang, M. et al. Broadband electro-optic frequency comb generation in a lithium niobate microring resonator. Nature 568, 373–377 (2019).

2. Secondly, even if mode mixing is unavoidable in the waveguide, its effect can be weakened by adding a cladding layer [Wang, Jingyi, et al. "Polarization coupling of X-cut thin film lithium niobate based

waveguides." IEEE Photonics Journal 12.3 (2020): 1-10] and cancelled out by the design of the radius of the bending waveguide [Cheng, Jian, et al. "Ultra-efficient second harmonic generation via mode phase matching in integrated lithium niobate racetrack resonators." Optics Express 31.22 (2023): 36736-36744]. It is worth noting that in this case the strength of mode mixing is insensitive to a larger or smaller radius, so that compactness can be ensured at the same time.

We agree that appropriate structural design can mitigate this issue in LNOI, and the theoretical study [1] mentioned by the reviewer is cited in our manuscript to illustrate this point.

[1] Wang, Jingyi, et al. "Polarization coupling of X-cut thin film lithium niobate based waveguides." IEEE Photonics Journal 12.3 (2020): 1-10.

3. At last, the view of birefringence in this work is quite one-sided. From another perspective, huge birefringence induced mode-mixing on X-cut LNOI can be used to develop the polarization-insensitive devices [Li, Xuepeng, et al. "Monolithic 1×4 Reconfigurable Electro-Optic Tunable Interleaver in Lithium Niobate Thin Film." IEEE Photonics Technology Letters 31.20 (2019): 1611-1614] that is highly desired in the optical communication networks.

We thank the reviewer for this comment. The polarization-insensitive device [1] mentioned by Reviewer #1 did not utilize the birefringence of x-cut LNOI. Instead, it employed a typical asymmetric Mach-Zehnder interferometer (AMZI) layout, where one arm of the AMZI featured two additional identical straight waveguides placed along the z-direction. As light propagated along the z-direction, both TE and TM modes utilized the ordinary refractive index, resulting in a polarization-independent performance of the device. This principle is applicable to any material platform. In particular, Xuepeng et al write: "Such a waveguide layout allows polarization-independent center wavelengths and channel spacing because no material birefringence emerges when light travels along the z direction" Therefore the example given by the reviewer here is not related to the material birefringence and can be applied also to materials that are optically isotropic. Indeed reading the report by Xuepeng et al. one may find that We also point out that polarization rotators and splitters, as well as polarization insensitive components, exist in isotropic optical materials such as silicon [2] and silicon nitride [3].

[1] Li, Xuepeng, et al. "Monolithic 1×4 Reconfigurable Electro-Optic Tunable Interleaver in Lithium Niobate Thin Film." IEEE Photonics Technology Letters 31.20 (2019): 1611-1614

[2] A, Stroganov, et al. "Efficient Polarization Rotator for Thick-film Si₃N₄ Integrated Photonics Platform" OSA Technical Digest (Optica Publishing Group, 2021), paper IM4A.5

[3] Z, Xiao, et al. "Ultra-compact low loss polarization insensitive silicon waveguide splitter"

In addition, huge-birefringence itself could also enable the high-efficiency and broadband nonlinear conversion in a poling-free configuration [5,6]. However, these applications are blocked for the LTOI due to the limited birefringence.

There are multiple methods to achieve phase matching. The huge birefringence can indeed be used for phase matching as in bulk optical crystals. That said, the most successful and most prevalent method to achieve phase matching is periodic poling – i.e. periodically poled waveguides (e.g. PPLN or PPLT). Therefore we do not believe that the absence of birefringence limits in any way the application scope of ferroelectrical photonic integrated circuits. For example, recent breakthroughs in generating e.g. few cycle squeezed vacuum [1] or mid-IR comb [2], all relied on **periodic poling**.

In contrast, birefringent phase matching mentioned by reviewer #1 is **rarely used in LN integrated photonics** because it precludes frequency conversion mediated by the largest component d_{33} (~27 pm/V), which is more than five times the second largest component d_{31} (~4.7 pm/V). For interactions

that make use of d_{33} (~27 pm/V), an optical field at all three frequencies involved must be polarized along the extraordinary axis, which cannot be achieved by birefringent phase matching. Actually, the mainstream approach for nonlinear conversion in LN and LT still uses the poling configuration to achieve quasi-phase-matching.

Even for the case of poling-free or poling-unavailable configuration, alternative phase matching mechanisms including mode phase matching (used in most cases) and natural phase matching widely exist. In fact, LT is often favored over LN as a material for nonlinear frequency conversion, in particular in the visible wavelength range, due to the higher optical damage threshold and lower photorefractive effect that is mediated by the larger optical bandgap.

[1] Nehra, Rajveer, et al. "Few-cycle vacuum squeezing in nanophotonics." *Science* 377.6612 (2022): 1333-1337.

[2] Roy, Arkadev, et al. "Visible-to-mid-IR tunable frequency comb in nanophotonics." *Nature Communications* 14.1 (2023): 6549.

Action taken: we outline in the manuscript the potential benefits of birefringence citing papers mentioned by the reviewer. We added the following sentence:

"Birefringence complicates the designs of compact PICs and is only useful in some special cases, such as birefringence phase matching."

4. Please see these references as follows to support the above comments.

References

[1] Boes, Andreas, et al. "Lithium niobate photonics: Unlocking the electromagnetic spectrum." *Science* 379.6627 (2023): eabj4396.

[2] Wang, Jingyi, et al. "Polarization coupling of X-cut thin film lithium niobate based waveguides." *IEEE Photonics Journal* 12.3 (2020): 1-10.

[3] Cheng, Jian, et al. "Ultra-efficient second harmonic generation via mode phase matching in integrated lithium niobate racetrack resonators." *Optics Express* 31.22 (2023): 36736-36744.

[4] Li, Xuepeng, et al. "Monolithic 1×4 Reconfigurable Electro-Optic Tunable Interleaver in Lithium Niobate Thin Film." *IEEE Photonics Technology Letters* 31.20 (2019): 1611-1614.

[5] Lu, Chuanyi, et al. "Highly tunable birefringent phase-matched second-harmonic generation in an angle-cut lithium niobate-on-insulator ridge waveguide." *Optics Letters* 47.5 (2022): 1081-1084.

[6] Lin, Jintian, et al. "Phase-Matched Second-Harmonic Generation in an On-Chip LiNbO₃ Microresonator." *Physical Review Applied* 6.1 (2016): 014002.

What's more, the authors emphasize that, Lithium Niobate wafers are costly and not available in high volumes. Actually, optical-grade LTOI are also costly according to what I know today, as for LiTaO₃ adopted for 5/6G wireless filters, LNOI also can be used as a SAW filters and also commercialized already; and the production of optical-grade LTOI wafers are no mass production, On the contrary, the price of LNOI is gradually decreasing.

As mentioned already in the earlier part of the reply, the prospects for LTOI, whether optical or acoustic grade, in terms of excellent performance, low cost, and mass production, are assured. We indeed demonstrated that acoustic grade can achieve almost the same performance and the optical grade is not required for our result. We showed in Extended Data Fig.1 that are fabricated with acoustic grade LTOI, the intrinsic loss rate is 42 MHz (F6, acoustic grade), comparable to 35 MHz (F3, optical grade).

As for the cost and production volume comparison between LTOI and LNOI, our team (Prof. Ou Xin) has over 20 years of experience in utilizing ion-cutting processes for the preparation of LNOI and LTOI,

and we have successfully industrialized the relevant products. Regarding the cost assessment, we provide a detailed response here:

- The reviewer emphasizes that what we used is optical grade LT. **However, the optical-grade LT we used does not increase the cost of LTOI wafers!** In order to increase the **transparency of the manuscript**, we also show the optical performance of LTOI prepared from the acoustic grade LT (see Extended Data Fig1). We had an explanation of the material difference between acoustic and optical LT in our manuscript, the only material difference lies in whether the material has undergone a chemical reduction process: acoustic grade LT is reduced, but optical grade LT is not. An additional reduction process is needed for acoustic LT to avoid the pyroelectric effect during filter fabrication. In fact, the cost of an LT wafer (optical grade) without the reduction process is even lower.
- The cost difference between LNOI and LTOI occurs during the film fabrication. The price of the bulk LN wafer is around 100 US dollars which is only 5% of the price of LNOI around 2000 US dollars, therefore, the efficiency of the ion implantation for ion-slicing is the bottleneck and decides the final cost of the LNOI, ion-slicing of LN can be only realized by the inert gas Helium ion implantation due to the LN-H bonding issue. However, similar to SOI fabrication, H⁺ implantation can be used for ion-slicing of LT. This is a significant difference since the current of an H ion beam is one order of magnitude higher than the Helium ion beam according to most commercial ion implanters, e.g., GSD 200E and CL-S400MB-H. This is due to the ionization efficiency of H being much higher than inert He gas. Meanwhile, for ion-slicing of the standard 500nm thin film, **180 keV He for LN** and **80 keV H for LT** are required respectively, since the He ion is heavier than the H ion. Additionally, existing machinery in the SOI industry can be used for LTOI preparation, which does not apply to LNOI preparation. Therefore, in contrast to the high cost of LNOI, **the cost of 6-inch LTOI is around 400 US dollars and approaching the cost of SOI wafers.** The continuous reduction in LTOI cost will be guaranteed by the further expansion of Smart-cut facilities, which is driven by the scale application in high-performance RF filters for cellular signals.

Figure R2. The beam current of an H⁺ ion beam is one order of magnitude higher than that of the He⁺ according to most commercial ion implanters.

- There is no doubt that the **production volume of LTOI exceeds that of LNOI by several orders of magnitude** driven by its widespread use in acoustic filters for 5G communications due to its superior temperature stability and low loss performance. The production volume of 150 mm LTOI (referred to as Piezo-OI in their report) is projected to reach 750k wafers per year by 2024 [2]. In contrast, the production volume of LNOI wafers is less than 10k wafers per year, estimated according to the revenue of the two main suppliers - NanoLN and NSIT (of which Prof. Xin Ou is a co-founder) from China. The huge market of LTOI in consumer electronics will provide significant cost and volume advantages for LT

photonics, similar to how Silicon Photonics benefits from the widespread applications of SOI in consumer electronics.

Action taken: We added that statement of the material and cost comparison of acoustic and optical LT in the manuscript.

“Note that both optical grade and acoustic grade LT wafers are congruent compositions and their cost is essentially the same. The difference between acoustic and optical LT only lies in whether the material has undergone a chemical reduction process: acoustic grade LT is reduced, but optical grade LT is not.”

We have revised the description of the ion implantation process in LTOI fabrication within the main text to highlight the higher fabrication efficiency achieved through the use of hydrogen ions, which require a lower implantation energy (100 keV) and have a higher beam current, compared to the traditional LNOI fabrication process that uses helium ion implantation with energies exceeding 200 kilo electronvolts.

“In contrast to the well-established LNOI preparation process that utilizes helium ion implantation with an energy higher than 200 keV, the fabrication of LTOI favors hydrogen ions with half the implanted energies 100 keV and beam current ten times higher, as found in most commercial ion implanters, simplifying the wafer production.”

Optical-grade LiTaO₃ wafer fabrication in the manuscript and the photonic devices fabricated in the LTOI firstly are novel, but the photonic devices are extensively researched in other materials platform and the experiment result in the manuscript not reflecting the author's claimed advantages

We are surprised by this comment and respectfully disagree with the criticism that the advantages of LTOI are not well-reflected. On the contrary, our results clearly reflect the advantages of LTOI over LNOI across many aspects – that have been extensively reported in the literature - including:

- mass production capabilities, due to application in 5G
- larger bandgap allowing PICs for near UV
- lower optical birefringence
- Raman suppression
- lower photorefractive effect, higher optical damage threshold

We performed additional measurements to compare the photorefractive effect in LNOI and LTOI. The measurement schematic is presented in Figure R3. We lock the CW pump laser to the cavity resonance, launch the constant power into the microresonator and measure the resonance shift over time. We use an LT microresonator (D101_A2_F2_C3_R202) with resonance at 193.638 THz with intrinsic cavity linewidth of $\kappa_0/2\pi = 18.8$ MHz and $\kappa_{ex}/2\pi = 132$ MHz (total linewidth 150.8 MHz). LN microresonator with the same design (D101_03_F2_C3_R202) has a resonance at 193.638 THz with intrinsic cavity linewidth of $\kappa_0/2\pi = 33.4$ MHz and $\kappa_{ex}/2\pi = 232$ MHz (total linewidth 265.4 MHz). In both experiments, we monitor the optical power on the bus waveguide (fiber to chip to fiber transmission is 10% for both chips). We set the optical power to 0.75 mW (at the bus waveguide). Measurement results are presented in Figure R4, the resonance shift is >5 times larger for the LN device than for the LT device. Given the higher total linewidth for the LN resonator and the much larger resonance shift, we conclude that the photorefractive effect is larger for LN than for LT. This observation is consistent with results obtained from measurements in bulk LT and LN crystals, as reported by Chen et al [1].

Figure R3. (a) Schematic for measurement of photorefractive effect in microresonators. (b) Resonance shift over time for LN and LT microresonators with similar FSRs and loaded cavity linewidth upon launching 0.75 mW on the bus waveguide.

[1] Chen, F. S. "Optically induced change of refractive indices in LiNbO₃ and LiTaO₃." Journal of applied physics 40.8 (1969): 3389-3396.

Action taken: we added new experimental data on the photorefractive comparison between LNOI and LTOI (Extended Data Figure 3), and provided detailed discussions in the Methods section "Photorefractive effect comparison between LT and LN".

Referee #2 (Remarks to the Author):

The manuscript has been revised very thoroughly and comprehensively, which has improved the quality of the content even further. The points I made in the first review have been fully addressed. The authors also address the criticism expressed by Reviewer #1 thoroughly and, in my opinion, appropriately and convincingly. Overall, I recommend the paper in its present form for acceptance by Nature.

We thank the referee for the careful study of the revised manuscript and positive recommendation.